# CADRILLE: MULTIMODAL CAD RECONSTRUCTION WITH REINFORCEMENT LEARNING

**Maksim Kolodiazhnyi**[1][*]    **Denis Tarasov**[2][*]    **Dmitrii Zhemchuzhnikov**[1]

**Alexander Nikulin**[1]    **Ilya Zisman**[3]    **Anna Vorontsova**    **Anton Konushin**[1]

**Vladislav Kurenkov**[3]    **Danila Rukhovich**[4][†]
[1]Lomonosov Moscow State University; [2]ETH Zurich; [3]Innopolis University;
[4]Institute of Mechanics, Armenia

## ABSTRACT

Computer-Aided Design (CAD) plays a central role in engineering and manufacturing, making it possible to create precise and editable 3D models. Using a variety of sensor or user-provided data as inputs for CAD reconstruction can democratize access to design applications. However, most existing methods focus on a single input modality: point clouds, images, or texts, which limits their generalizability and robustness, while few multimodal approaches struggle to deliver competitive quality. Leveraging advances in vision-language models (VLM), we propose `cadrille`, a multimodal CAD reconstruction model that takes inputs of three modalities and outputs executable Python code for CAD reconstruction. Inspired by large language model (LLM) training paradigm, we adopt a two-stage pipeline: supervised fine-tuning (SFT) on large-scale procedurally generated data, followed by reinforcement learning (RL) fine-tuning using online feedback, obtained programatically. On the DeepCAD benchmark, our SFT model outperforms existing single-modal approaches in all three input modalities simultaneously. More importantly, after RL fine-tuning, `cadrille` sets new state-of-the-art on as many as 10 benchmarks across three modalities and four datasets, including a real-world one. Code is avaliable at `https://github.com/col14m/cadrille`.

## 1 INTRODUCTION

Computer-Aided Design (CAD) is the core of modern engineering and manufacturing, providing the tools to create detailed and modifiable 3D models (Briere-Cote et al., 2012). Creating CAD models manually requires skills, time, and effort. To simplify this process, CAD reconstruction aims at generating CAD models directly from scanned objects, making the process faster, cheaper, and overall more accessible (Rukhovich et al., 2024).

Typically, CAD models are created with a sequence of 2D sketches and 3D operations (Willis et al., 2021; Wu et al., 2021). This representation allows CAD models to be easily edited, making it prevalent in popular CAD tools like SolidWorks and AutoCAD and in CAD generation research. Most existing CAD generation methods define CAD sequences using special command tokens (Khan et al., 2024a; Wu et al., 2021). However, state-of-the-art results are obtained via mapping CAD sequences to casual Python code (Rukhovich et al., 2024). Following the same paradigm, we generate CAD models as executable Python scripts.

The most well-studied input modality in CAD reconstruction is naturally a point cloud (Rukhovich et al., 2024). However, point clouds can only be obtained when a physical 3D object is available, while scanning usually requires special equipment, making the process complicated for non-experts. Images capture finer details and can be sourced using customer low-end devices (e.g., smartphone

---

[*]Equal contribution
[†]Corresponding author: rukhovich@mechins.sci.am

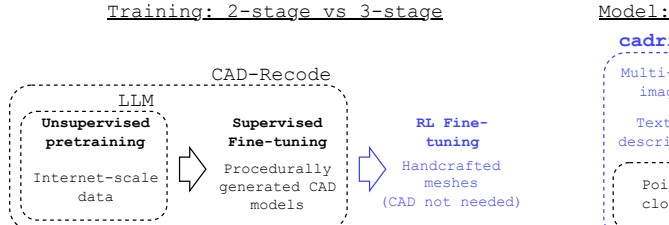 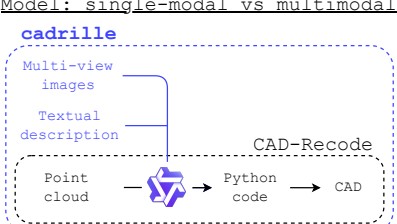

Figure 1: Compared to state-of-the-art CAD-Recode, the only existing method that converts point clouds into Python code, `cadrille` has two key novelties. First, it goes beyond the standard training scheme and adapts LLM RL fine-tuning for CAD reconstruction (left). Moreover, besides point clouds only accepted by single-modal CAD-Recode, `cadrille` extends to images and textual descriptions, making it the first multimodal approach delivering state-of-art results (right).

cameras), hence relaxing the hardware requirements (Chen et al., 2025; 2024). In the meantime, textual descriptions can enrich the object representation with semantic context (Khan et al., 2024b). Using various input modalities, such as multi-view images, or natural language descriptions, would make design assistance applications simple even for non-experienced users. Recent emergence of vision-language models provides a solid ground for multimodal CAD reconstruction. However, the first multimodal methods in that vein (Xu et al., 2024b; Wang et al., 2025b) are dramatically inferior to single-modal approaches, so the full potential of VLMs for CAD reconstruction is yet to be unleashed.

Existing CAD reconstruction methods face generalization issues due to how they are trained. Specifically, handcrafted CAD datasets are small and limited in diversity (Khan et al., 2024a), while models trained with procedurally generated data struggle to transfer to the real-world domain (Rukhovich et al., 2024). In the vein of the standard LLM training pipelines (Shao et al., 2024), Chen et al. (2025); Wang et al. (2025a); Guan et al. (2025) bring RL fine-tuning into the CAD reconstruction context. However, all of them perform both supervised and RL fine-tuning on the same dataset, which does not help bridging the gap between training and testing data. In contrast, we use voluminous procedurally generated data for supervised training, while valuable but scarcer handcrafted data is reserved for RL fine-tuning. This scheme eliminates the need for large-scale handcrafted data and allows the model to first generalize across the CAD domain and then specialize using preference-based objectives.

Our experiments show that `cadrille` outperforms existing modality-specific baselines in accuracy. Moreover, RL fine-tuning ensures validity of generated Python code, which posed a challenge for prior works. As a result, the proposed approach demonstrates impressive robustness and sets a new state-of-the-art on several CAD datasets, including a real-world CC3D (Mallis et al., 2023). Essentially, this opens up new possibilities for generalization in open-world scenarios.

In summary, our contributions are as follows:

- We present `cadrille`, an LLM-based model able to process point clouds, images, and textual inputs, and generate Python scripts for CAD reconstruction;
- We are the first to prove that RL fine-tuning improves multimodal CAD reconstruction;
- With a single model, we simultaneously achieve state-of-the-art results across three input modalities (point clouds, images, texts) and four datasets (DeepCAD, Fusion360, CC3D, Omni-CAD), a total of 10 benchmarks, making it the most comprehensive evaluation of CAD reconstruction methods up-to-date.

## 2 RELATED WORK

**CAD Generation**  Existing CAD generation methods can be classified into three categories based on CAD model representations: constructive solid geometry (CSG) (Du et al., 2018; Sharma et al., 2018; Nandi et al., 2018; Ellis et al., 2019; Tian et al., 2019; Friedrich et al., 2019; Kania et al., 2020; Ren et al., 2021; Yu et al., 2022; 2023), boundary representation (B-rep) (Wang et al., 2020; Sharma

et al., 2020; Lambourne et al., 2021; Wang et al., 2022; Guo et al., 2022; Jayaraman et al., 2023; Liu et al., 2024a;b; Xu et al., 2024c; Li et al., 2025b) and CAD sequence (Wu et al., 2021; Lambourne et al., 2022; Ren et al., 2022; Xu et al., 2022; 2023; Zhang et al., 2025b; Badagabettu et al., 2024; Chen et al., 2024; Khan et al., 2024a;b; Ma et al., 2024; Mallis et al., 2024; Li et al., 2025a; Doris et al., 2025; He et al., 2025; Yuan et al., 2025; Wang et al., 2025b). In the CSG (Foley, 1996) paradigm, CAD is represented as a CSG tree constructed using boolean operations (union, subtraction, difference) of geometric primitives (e.g., cubes, cylinders, or spheres). This approach fails to express intricate shapes and is generally not well-aligned with how engineers and designers actually build CAD models. B-rep (Ansaldi et al., 1985) is a graph that describes connections between faces, edges, and vertices of a 3D model. Creating a B-Rep requires enforcing topological consistency on edges, which introduces additional complexity to the generation procedure and complicates editing of generated models.

**CAD Sequence Reconstruction**     Unlike general CAD generation, which may prioritize plausibility, diversity, or creativity in design, CAD reconstruction aims at faithfulness to the given inputs, requiring the output model to match the original shape.

Point clouds are the most well-studied input modality in CAD reconstruction. The seminal work on point cloud-based CAD reconstruction by Wu et al. (2021) proposed encoding CAD sketch-and-extrude sequences as special tokens. Beyond that, DeepCAD, a large-scale dataset of 180k hand-crafted CAD models, was presented. Subsequent works (Dupont et al., 2024; Khan et al., 2024a; Xu et al., 2023) adopted the same CAD representation and trained on the same DeepCAD dataset. More recently, CAD-Recode (Rukhovich et al., 2024) introduced a paradigm shift by representing CAD models as Python code, providing greater expressiveness and flexibility, and released a new dataset of approx. 1 million procedurally generated CAD models.

Only few recent works (Chen et al., 2024; Khan et al., 2024b; Wang et al., 2025c; Chen et al., 2025) have explored CAD reconstruction from alternative input modalities, such as single- or multi-view images and natural language descriptions. These approaches extend the DeepCAD dataset by rendering synthetic views or generating textual captions for existing CAD models. Among them, CADCrafter (Chen et al., 2025) stands out for its unified framework that handles both single- and multi-view inputs, whether rendered or real. The seminal Text2CAD (Khan et al., 2024b) uses a vision-language model (VLM) to generate detailed captions for DeepCAD shapes and then trains a model to predict CAD sequences from those textual descriptions. Its recent follow-ups (Xie & Ju, 2025; Govindarajan et al., 2025; Guan et al., 2025; Wang et al., 2025a) adapted large language models for text-based CAD reconstruction.

Generally, state-of-the-art CAD reconstruction approaches are tailored to process specific input modalities with distinct architectures, while multimodal CAD reconstruction remains relatively underexplored. Recent CAD-GPT (Wang et al., 2025b) predicts a CAD model given a single image and textual description, while CAD-MLLM (Xu et al., 2024b) pioneers three-modal CAD reconstruction, yet both these methods fall behind single-modal state-of-the-art results (Rukhovich et al., 2024; Khan et al., 2024b; Chen et al., 2025) by a large margin. This makes our `cadrille` the first multimodal CAD reconstruction approach handling point clouds, images, and texts within a unified framework, that outperforms single-modal top-performing methods.

**RL for CAD Reconstruction**     Reinforcement learning is used for CAD reconstruction from images (Sharma et al., 2018; Chen et al., 2025; Zhang et al., 2025a), and from B-Rep (Yin et al., 2025). Recent LLM-based CADFusion (Wang et al., 2025a) and CAD-Coder (Guan et al., 2025) address CAD reconstruction from texts, both performing supervised and RL fine-tuning on the same Deep-CAD dataset. On the contrary, we investigate RL fine-tuning for multimodal CAD reconstruction and improve the reconstruction quality by using large-scale procedurally generated data for SFT.

## 3    CAD SEQUENCE RECONSTRUCTION

**Problem Formulation**     The task of CAD reconstruction implies recovering a CAD model given a multimodal input $q$, which can be a 3D point cloud, a set of images, or a textual description. We represent CAD models as Python scripts (Rukhovich et al., 2024) that, when executed, generate a parametric Boundary Representation (B-Rep) of a 3D shape. Respectively, given an input $q$, we

search for a trainable policy $\pi_\theta$, s. t. $\pi_\theta(q)$ produces a token sequence $\tau$, which is essentially a text of a Python program generating a CAD model.

**Multimodal Data** For training a model, we derive all input modalities from ground-truth CAD models (Fig. 2). Below, we describe how each modality is constructed according to the established data generation protocols (Chen et al., 2025; Khan et al., 2024b; Wu et al., 2021).

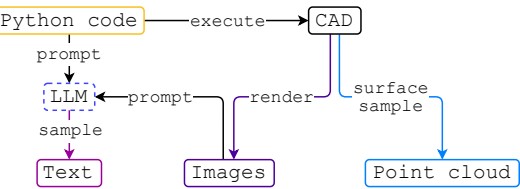

Figure 2: Overview of multimodal data generation pipeline producing textual descriptions, multi-view images and point clouds.

Given a CAD model as a parametric 3D shape (B-Rep), we sample points directly from the parametric surfaces of the model. Modern CAD engines provide built-in routines for surface sampling, making it simple and straightforward.

To generate images, the B-Rep is first tessellated, i.e., converted into a triangle mesh that approximates the surface geometry. Then, this mesh can be rendered from multiple viewpoints to produce multi-view image inputs.

Generating textual data is notably more challenging. Since our goal is accurate geometry reconstruction rather than generating a semantically relevant sample, inputs should provide detailed and comprehensive geometric information. Consequently, loose textual descriptions are generally insufficient. The necessary level of granularity is investigated in Text2CAD (Khan et al., 2024b), where LLMs and VLMs are combined in a multi-stage sophisticated pipeline that generates textual descriptions from both the CAD sequence and rendered images.

## 4 PROPOSED METHOD

### 4.1 CADRILLE ARCHITECTURE

The `cadrille` architecture is depicted in Fig. 3. The model accepts inputs in the form of a point cloud, a set of images, or a text prompt, and outputs a Python code, that, when executed, produces a CAD model. `cadrille` is build on top of a VLM that natively supports text and image inputs and is already capable of generating Python code. Textual input is passed through the original embedding layer, and images are processed with an original visual encoder. The point cloud processing logic is the same as in CAD-Recode. Specifically, we use a single projection layer to embed 3D points, sample points from the surface via furthest point sampling, and do not use normals.

### 4.2 SUPERVISED FINE-TUNING

As shown on Fig. 1, `cadrille` benefits from three stages of training. First, we use VLM which is pre-trained on the internet-scale data in the unsupervised manner. After this stage, VLM is able to process textual and visual inputs and generate Python code, but lacks mechanisms to handle point clouds. In this work, we do not perform any unsupervised VLM training, but enjoy the capabilities of an already trained model.

The second stage is supervised fine-tuning for a specific task. During SFT, a model develops the ability to process point clouds and learns a policy $\pi_\theta$ to map multimodal inputs $q$ to Python codes $\tau$, making SFT an essential part of `cadrille` pipeline. We construct a training dataset $\mathcal{D}$ of samples $(q, \tau)$, where $q$ is a multimodal input. The training procedure aims to minimize cross-entropy between ground truth and predicted Python code tokens:

$$\mathbb{E}_{(q,\tau)\sim\mathcal{D}}\left[\log \pi_\theta(\tau \mid q)\right]$$

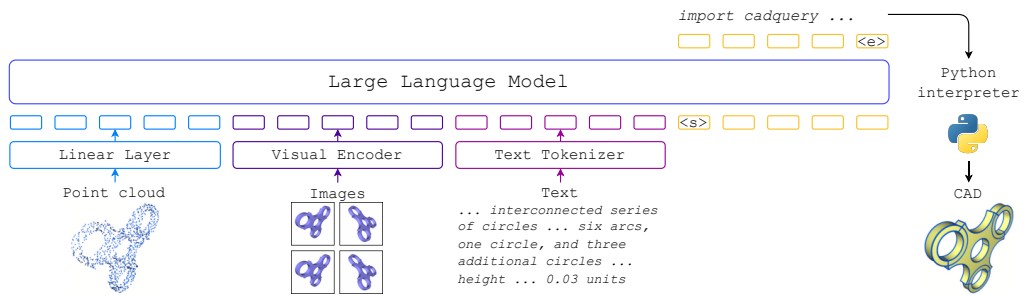

Figure 3: Overview of `cadrille`. It can handle three input modalities within a unified framework. Point clouds are processed with a trainable projection layer, while images and texts are passed to a VLM directly. The output of the model is an executable Python script for CAD generation.

### 4.3 LIMITATIONS OF SFT

Two-stage training has already been adopted in CAD-Recode, which employs supervised fine-tuning (SFT) to adapt a pretrained language model for point cloud-based CAD reconstruction. However, this strategy reveals its limitations in a cross-domain scenario: CC3D IoU is as low as 60% and the invalidity ratio (IR) almost reaches 10%, which means that every tenth prediction fails to produce a valid output (Tab. 3, row 2). To mitigate this issue, CAD-Recode uses a test-time sampling technique. For each input query, 10 candidate Python programs are generated, and the candidate with the highest IoU is selected. After that, IoU increases to 74%, while IR drops below 0.5%. However, this improvement comes at the cost of a 10x increase in inference time. Can similar gains be achieved without sacrificing test-time efficiency?

To maintain fast and simple inference, we shift our focus to improving the training process. Training solely on procedurally generated CAD data might limit performance in real-world applications. Nevertheless, training on handcrafted models also presents challenges, e.g., Rukhovich et al. (2024) shows that SFT directly on the DeepCAD dataset harms performance, leading to a 10% drop in IoU.

Our experiments confirm that simply mixing procedurally generated and handcrafted data for training fails to improve results and can even degrade performance (Tab. 3, row 4). We attribute this to inconsistency in CAD sequences across datasets: for instance, DeepCAD models are constructed using commands like extruded cuts and symmetric extrusions, which are not present in the generation procedure of the CAD-Recode dataset.

To address this limitation, we introduce a novel third stage in the training pipeline, namely, reinforcement learning fine-tuning on handcrafted data not annotated with CAD sequences. This approach resolves inconsistency issues while still allowing the model to adapt to real-world domain.

### 4.4 RL FINE-TUNING

We formulate RL fine-tuning as follows. Given a dataset of inputs (either images or point clouds) $\mathcal{D} = \{q_i\}_{i=1}^{N}$, and reward function $R(\tau)$, we learn LLM policy $\pi_\theta(\tau \mid q)$ that generates a Python code $\tau$ for an input $q$, s.t. it maximizes the expected reward $\mathbb{E}_{q_i \sim D, \tau_i \sim \pi_\theta(\cdot|q_i)} [R(\tau_i)]$.

Note that at this stage annotated pairs of $(q, \tau)$ are not needed for supervision, since Python codes $\tau$ are being sampled from the trained SFT model. In fact, **CAD sequences are not needed for RL fine-tuning**, and the data requirements can be relaxed to 3D meshes instead. This is especially beneficial from the practical perspective, as RL fine-tuning can be performed using generally more accessible mesh datasets, which opens new possibilities for training models accommodated to artifacts present in real-world data.

The reward function $R(\tau)$ is a combination of terms that address precision and robustness:

$$R(\tau) = r_{\text{IoU}}(\tau) + r_{\text{invalid}}(\tau),$$

where $r_{\text{IoU}}$ is an IoU between the CAD model produced by $\tau$ and ground truth 3D mesh, additionally multiplied by a factor of 10 to enforce precise reconstruction. $r_{\text{invalid}}$ penalizes invalid predictions: it is set to -10 for invalid $\tau$ and 0 otherwise.

Empirically, we found that hard example mining leads to a faster convergence of RL fine-tuning. Consequently, we only use examples $q$ where the reward $R(\tau)$ averaged over three samples produced by the SFT model is less than $R_{\text{th}}$, where $R_{\text{th}} = 7.5$.

**DPO**  Direct Preference Optimization (DPO) Rafailov et al. (2023) learns from pairwise preference data, approximating an implicit reward via a reparameterized Bradley-Terry model.

We construct the training dataset by sampling $K = 5$ Python codes $\tau$ for each input $q$ from the SFT model $\pi_{\theta_r}$. At each training step for the given sample, we randomly select two outputs. The output with a larger reward $R(\tau)$ is considered to be a preferred prediction $\tau_w$, and another is non-preferred $\tau_l$. The optimization objective is formulated as:

$$\mathbb{E}_{(q,\tau_w,\tau_l)\sim\mathcal{D}} \left[ \log \sigma \left( \beta \log \frac{\pi_{\theta_t}(\tau_w \mid q)}{\pi_{\theta_r}(\tau_w \mid q)} - \beta \log \frac{\pi_{\theta_t}(\tau_l \mid q)}{\pi_{\theta_r}(\tau_l \mid q)} \right) \right]$$

DPO training starts with $\pi_{\theta_r}$ and proceeds for 10 epochs. After that, the SFT model is replaced with the latest $\pi_{\theta_t}$, and trained for another 10 epochs. In this way, the model gradually diverges from the original SFT model. In our experiments, we found it to be beneficial for performance.

However, DPO performance is upper-bounded by the quality of the best generated sample for a given example. This limitation cannot be overcome without generating additional samples, so we adapt an online RL approach that can benefit from newly generated samples.

**Dr. CPPO**  We combine two recent modifications of the GRPO: Dr. GRPO Liu et al. (2025b) which eliminates the need for a reference model and modifies the objective, and CPPO Lin et al. (2025) which uses samples with the strongest signal. The hybrid approach ensures both computational efficiency and accuracy; hereinafter, it is referred to as Dr. CPPO.

$G$ sequences $\{\tau_g\}_{g=1}^{G}$ are sampled from the current policy $\pi_{\theta_{\text{old}}}(\tau \mid q)$ for a given input $q$ with temperature $T = 1.0$. For each output $g$, the advantage $A_g$ is estimated as $A_g = r_g - \text{mean}(\{r_i\}_{i=1}^{G})$. $N$ samples with the highest $|A_g|$ are used to form a batch $\mathcal{B}$ and perform policy update by maximizing PPO Schulman et al. (2017) objective:

$$\mathbb{E}_{\{\tau_g\}\sim\mathcal{B}} \left[ \min \left( \frac{\pi_{\theta_t}(\tau_g \mid q)}{\pi_{\theta_{\text{old}}}(\tau_g \mid q)} A_g, \ \text{clip} \left( \frac{\pi_{\theta_t}(\tau_g \mid q)}{\pi_{\theta_{\text{old}}}(\tau_g \mid q)}, \ 1 - \epsilon, \ 1 + \epsilon \right) A_g \right) \right]$$

## 5 EXPERIMENTS

**Datasets**  DeepCAD (Wu et al., 2021) (denoted as D in Tables) serves as our primary benchmark for supervised training. We adopt the Text2CAD version of DeepCAD, which enriches it with textual descriptions. The train set comprises approximately 160k samples, while 8046 are left for testing.

For SFT, we also use the procedurally generated CAD-Recode (Rukhovich et al., 2024) dataset (denoted as R in the tables). It is an order of magnitude larger than DeepCAD, consisting of approximately 1 million CAD programs written in CadQuery (Authors, 2024), a parametric Python-based CAD language.

Fusion360 (Willis et al., 2021) (denoted as F) is a small CAD reconstruction benchmark with complex and realistic CAD models. In the standard evaluation protocol, only the test split (1725 samples) is used, as absence of Python CAD sequences makes it unsuitable for conventional supervised training. Still, we can use its train set (6900 samples) for our annotation-free RL fine-tuning.

To show the versatility and applicability of our approach, in addition to handcrafted and procedurally generated meshes, we report metrics on the real-world CC3D dataset (Mallis et al., 2023) (denoted as C). It contains 2973 point clouds sampled from scans of CAD models with noisy values, missing parts, and smoothed edges.

Omni-CAD (denoted as O) presented in the CAD-MLLM paper (Xu et al., 2024b) is a large-scale dataset of handcrafted CAD models, sourced from the web. We evaluate on its test split composed of over 27K samples, and report results in Appendix B.

**Metrics** Following CAD-Recode, we normalize ground truth CAD models so that they fit into $[-0.5, 0.5]^3$ and report three metrics: Chamfer Distance (CD), Intersection over Union (IoU), and Invalidity Ratio (IR). Since invalid CAD models introduce a notable bias into mean estimates, we report *median* CD computed using 8192 points. CD values are multiplied by $10^3$. The divergence between ground truth and reconstructed meshes is measured using IoU (in %). The IR indicates the percentage of generated sequences that do not produce a valid CAD model.

## 5.1 SUPERVISED FINE-TUNING

**Results on DeepCAD** In Tab. 1, we compare `cadrille` with single-modal CAD reconstruction methods on DeepCAD. Here, input modalities are denoted with subscripts: $p$ stands for point clouds, $i$ for images, and $t$ for texts. `cadrille` trained jointly on point clouds, multi-view images, and texts from the DeepCAD training set ($D_{pit}$) outperforms the modality-specific baselines. Noticeably, IR is reduced almost twice for point clouds (from 1.1 to 0.4) and 7x for images (3.6 to 0.5).

Training using the large-scale procedurally generated CAD-Recode dataset (R) consistently improves accuracy over training on the DeepCAD dataset. Since we also use a Qwen LLM model as in CAD-Recode (Qwen2-VL-2B against Qwen2-1.5B), comparable quality of point cloud-based reconstruction is expected. When training `cadrille` on point clouds and images ($R_{pi}$), it maintains the same accuracy on point clouds but additionally extends to images. After training with point clouds, images, and texts ($S_{pi} + D_i$), `cadrille` generalizes across modalities without loss of quality on each modality. For fair comparison, we do not apply any RL techniques in this series of experiments, and mix up training datasets trivially for SFT.

**Results on Fusion360 and CC3D** Both Fusion360 and CC3D datasets do not provide annotations in a compatible format, and are only used for testing in the standard evaluation protocol (Khan et al., 2024a). Accordingly, testing on these datasets is performed in a zero-shot scenario, which allows assessing the generalization ability of CAD reconstruction approaches. Furthermore, since CC3D contains real scans of objects, this experiment emulates real-world application.

We report CAD reconstruction quality from images and point clouds in Tab. 2 and 3, respectively. CADCrafter is the only method performing CAD reconstruction based on multi-view images. However, the authors of CADCrafter only report metrics on the DeepCAD dataset, and benchmarking it on other datasets is problematic since the code has not been released. To establish a baseline in image-based CAD reconstruction, we combine two off-the-shelf state-of-the-art methods, namely, multi-view reconstruction method LRM Hong et al. (2024); Xu et al. (2024a) and CAD-Recode. LRM takes multi-view images as inputs and produces a mesh, which is turned into a point cloud via surface sampling, and this point cloud is then passed to CAD-Recode to create a CAD model. As can be seen in Tab. 2, `cadrille` trained on CAD-Recode outperforms both baselines.

In Tab. 3, we compare `cadrille` against the state-of-the-art approaches originally trained on the DeepCAD (CAD-SIGNet) and CAD-Recode datasets. As could be expected, `cadrille` is on par with CAD-Recode, while delivering substantially better quality w.r.t. CAD-SIGNet.

## 5.2 REINFORCEMENT LEARNING

**RL on single modality boosts other modalities** In Tab. 2, we report accuracy of image-based CAD reconstruction on three benchmarks. Respectively, we fine-tune `cadrille` with images from DeepCAD and Fusion360 datasets. It is worth noticing that while Fusion360 cannot be used for direct supervised training, it can still contribute to RL fine-tuning where CAD sequences are not required, so we can benefit from adding it to the mixture. We denote DeepCAD and Fusion360 datasets without CAD sequence annotations as $D^-$ and $F^-$.

Surprisingly, RL fine-tuning on images appears to be beneficial for other modalities: as reported in Tab. 3, the model tuned on $D_i^- + F_i^-$ (row 6) delivers state-of-the-art quality of CAD reconstruction from point clouds as well.

| Method | Train Data | Point Cloud | | | Multi-view Images | | | Text | | |
|---|---|---|---|---|---|---|---|---|---|---|
| | | CD↓ | IoU↑ | IR↓ | CD↓ | IoU↑ | IR↓ | CD↓ | IoU↑ | IR↓ |
| PointNet→DeepCAD | $D_p$ | 9.64 | 46.7 | 7.1 | | | | | | |
| Point-BERT→HNC-CAD | $D_p$ | 8.64 | 65.3 | 5.6 | | | | | | |
| MultiCAD | $D_p$ | 8.09 | | 11.5 | | | | | | |
| TransCAD | $D_p$ | 4.51 | 65.5 | _1.1_ | | | | | | |
| PrismCAD | $D_p$ | 4.28 | 72.1 | 16.2 | | | | | | |
| Point2Cyl | $D_p$ | 4.27 | 73.8 | 3.9 | | | | | | |
| CAD-Diffuser | $D_p$ | 3.02 | 74.3 | 1.5 | | | | | | |
| CAD-SIGNet | $D_p$ | _0.29_ | _77.3_ | 5.0 | | | | | | |
| DINOv2→HNC-CAD | $D_i$ | | | | 2.08 | | 10.1 | | | |
| DINOv2→DeepCAD | $D_i$ | | | | 1.13 | | 10.6 | | | |
| CADCrafter | $D_i$ | | | | _0.26_ | | _3.6_ | | | |
| BERT→DeepCAD | $D_t$ | | | | | | | 32.82 | | 10.0 |
| CADmium | $D_t$ | | | | | | | 0.38 | | 4.3 |
| Text2CAD | $D_t$ | | | | | | | 0.37 | _71.5_ | 3.7 |
| CAD-Coder | $D_t$ | | | | | | | 0.33 | | 5.3 |
| Text-to-CadQuery | $D_t$ | | | | | | | _0.22_ | | **1.3** |
| **cadrille** | $D_{pit}$ | **0.25** | **79.4** | **0.4** | **0.25** | **78.2** | **0.5** | **0.21** | **81.1** | _1.4_ |
| CAD-Recode | $R_p$ | 0.18 | 87.1 | 3.1 | | | | | | |
| **cadrille** | $R_{pi}$ | 0.18 | 87.1 | 2.1 | 0.18 | 86.1 | 1.5 | | | |
| **cadrille** | $R_{pi}+D_t$ | 0.18 | 87.1 | 2.1 | 0.18 | 86.1 | 1.5 | 0.20 | 82.1 | 1.4 |

Table 1: Results on DeepCAD test set. The best results are **bold**, the second best are underlined. Our cadrille trained jointly on three modalities outperforms all existing modality-specific methods. Here, we report metrics obtained *without* RL fine-tuning or test-time sampling for fair comparison.

| Method | RL | Train Data | | DeepCAD | | | Fusion360 | | | CC3D | | |
|---|---|---|---|---|---|---|---|---|---|---|---|---|
| | | SFT | RL | CD↓ | IoU↑ | IR↓ | CD↓ | IoU↑ | IR↓ | CD↓ | IoU↑ | IR↓ |
| LRM→CAD-Recode | ✗ | $R_p$ | ✗ | 0.53 | 69.8 | 14.3 | 0.62 | 62.5 | 18.7 | 1.19 | 50.1 | 20.1 |
| CADCrafter | DPO | $D_i$ | $D_i$ | 0.26 | | 3.6 | | | | | | |
| **cadrille** | ✗ | $R_{pi}$ | ✗ | 0.18 | 86.1 | 1.5 | 0.20 | 77.6 | 3.2 | 0.81 | 56.1 | 7.7 |
| **cadrille** | ✗ | $R_{pi}+D_{pi}$ | ✗ | 0.19 | 85.6 | 0.6 | 0.23 | 75.2 | 2.6 | 1.17 | 53.1 | 6.0 |
| **cadrille** | DPO | $R_{pi}$ | $D_i^-+F_i^-$ | 0.18 | 86.9 | 1.8 | 0.20 | 78.5 | 1.7 | 0.89 | 56.0 | 3.9 |
| **cadrille** | Dr. CPPO | $R_{pi}$ | $D_i^-+F_i^-$ | **0.17** | **92.2** | **0.0** | **0.17** | **84.6** | **0.0** | **0.57** | **65.0** | **0.1** |

Table 2: Results of CAD reconstruction from multi-view images. With RL fine-tuning, cadrille achieves best results across three benchmarks.

**RL improves metrics in cross-dataset scenario** RL fine-tuning with DeepCAD and Fusion360 boosts accuracy on the test splits of the respective datasets. Yet, the performance gain is not limited to the domains seen by a model during SFT and RL fine-tuning. In image-based CAD reconstruction, CD is reduced from 0.81 to 0.57, while IR dropped dramatically from 7.7 to 0.1 (rows 3 and 6, respectively). When testing on point clouds, RL also improves all scores on CC3D, making IR less than 0.2%, which is negligible.

**Online RL outperforms offline RL** Fine-tuning cadrille using offline DPO reduces IR twice in most cases, while accuracy scores are not affected (rows 3 and 5 in both Tables). In the meantime, Dr. CPPO beats SFT in terms of all metrics, adding 3-9% to IoU scores and bringing IR under 0.2%

| Method | RL | Train Data | | DeepCAD | | | Fusion360 | | | Real-world CC3D | | |
|---|---|---|---|---|---|---|---|---|---|---|---|---|
| | | SFT | RL | CD↓ | IoU↑ | IR↓ | CD↓ | IoU↑ | IR↓ | CD↓ | IoU↑ | IR↓ |
| CAD-SIGNet | ✗ | $D_p$ | ✗ | 0.29 | 77.3 | 5.0 | 0.70 | 58.4 | 9.3 | 4.42 | 39.1 | 15.5 |
| CAD-Recode | ✗ | $R_p$ | ✗ | 0.18 | 87.1 | 3.1 | 0.19 | 79.1 | 5.0 | 0.54 | 60.5 | 9.8 |
| **cadrille** | ✗ | $R_{pi}$ | ✗ | 0.18 | 87.1 | 2.1 | 0.19 | 79.8 | 2.8 | 0.54 | 61.8 | 5.9 |
| **cadrille** | ✗ | $R_{pi}+D_{pi}$ | ✗ | 0.19 | 86.6 | 0.9 | 0.22 | 76.5 | 2.0 | 0.79 | 58.7 | 4.1 |
| **cadrille** | DPO | $R_{pi}$ | $D_i^-+F_i^-$ | 0.18 | 88.1 | 0.7 | 0.19 | 80.9 | 1.3 | 0.54 | 61.3 | 2.6 |
| **cadrille** | Dr. CPPO | $R_{pi}$ | $D_i^-+F_i^-$ | **0.17** | **90.2** | **0.0** | **0.17** | **85.0** | **0.2** | **0.47** | **67.9** | **0.2** |

Table 3: Results of CAD reconstruction from point clouds. cadrille performs on par with CAD-Recode when trained on the CAD-Recode dataset (R). With RL, cadrille establishes state-of-the-art on DeepCAD, Fusion360 and real-world CC3D.

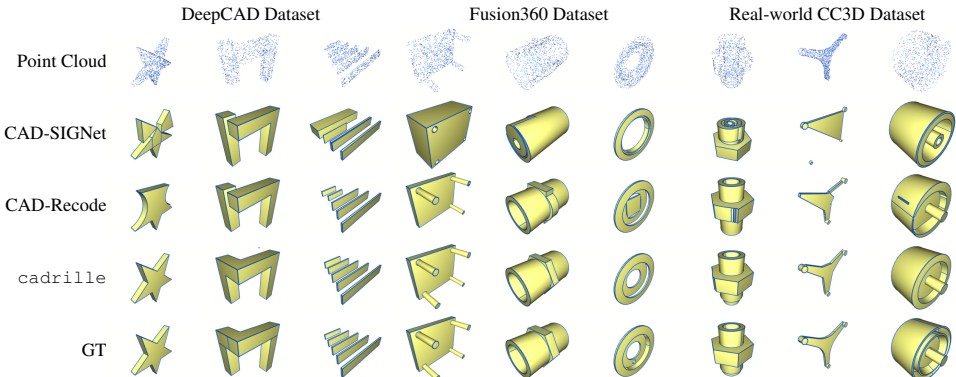

Figure 4: CAD models reconstructed from point clouds from the DeepCAD, Fusion360, and CC3D datasets.

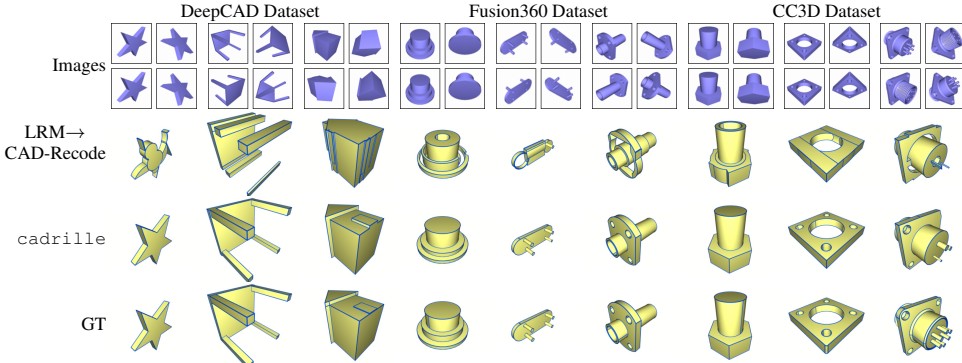

Figure 5: CAD models reconstructed from multi-view images on the DeepCAD, Fusion360, and CC3D datasets.

on all benchmarks (row 6). The observed improvement of CAD reconstruction accuracy aligns well with the experimental results obtained in other tasks where feedback can be programmatically computed Shao et al. (2024).

**RL fine-tuning beats SFT on a mixture** A common assumption is that mixing datasets improves generalization by increasing data diversity and volume. However, our experiments show that SFT with a plain mixture of CAD-Recode and DeepCAD datasets ($R_{pi}+D_{pi}$, row 4) does not lead to performance gains, and can even degrade results w.r.t. SFT with $R_{pi}$ (row 3). We attribute this effect to the domain gap between datasets, specifically, some CAD operations present in DeepCAD (e.g., symmetric extrusion, extruded cut) are lacking from CAD-Recode.

**Qualitative results** CAD models obtained with RL fine-tuning are depicted in Fig. 4 (from point clouds) and Fig. 5 (from multi-view images). Compared to predecessors, `cadrille` produces more geometrically plausible reconstructions and better restores fine details.

## 6 CONCLUSION

We introduced `cadrille`, a multimodal CAD reconstruction model that is capable of processing point clouds, multi-view images, and text inputs within a unified VLM-based framework. By adopting a two-stage training paradigm, namely, supervised fine-tuning on synthetic data followed by reinforcement learning fine-tuning with programmatic feedback, we improved both reconstruction quality and validity ratio. Our empirical study demonstrated that online RL approaches are especially beneficial in the CAD reconstruction scenario. `cadrille` achieves new state-of-the-art results in 10 CAD reconstruction benchmarks, including a real-world dataset, highlighting its

robustness, generalizability, and potential for further use in applications. Based on our study, we identify the following promising research directions for the future work: 1) combine modalities in one prompt to compensate for low-quality or missing inputs 2) perform RL fine-tuning on point clouds, and 3) increase complexity of procedurally generated data and volume of RL fine-tuning data to better adapt to real-world scans.

This work was supported by the The Ministry of Economic Development of the Russian Federation in accordance with the subsidy agreement (agreement identifier 000000C313925P4H0002; grant No 139-15-2025-012).

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

## A    QUALITATIVE RESULTS

**Text-based CAD reconstruction**    Fig. 6 shows results of text-based CAD reconstruction on the DeepCAD dataset. Long textual descriptions still cannot define even simple 3D shapes comprehensively and unambiguously, making CAD reconstruction from textual inputs the most challenging. Both Text2CAD and `cadrille` struggle to recover correct geometry, yet our approach yields more accurate predictions, which is also reflected in the quantitative metrics.

**Real-world single-image experiment**    Since our `cadrille` is trained only on synthetic renders, sim-to-real transfer can be eligibly questioned. To address the potential concerns about the applicability of our approach, apart from validating on real scans from the CC3D dataset, we also experiment with CAD reconstruction from a single real-world image.

The pipeline consists of three steps. First, an input image is processed using the recent image-to-mesh InstantMesh Xu et al. (2024a) method, that produces a mesh. Second, we follow the same protocol as in other experiments to convert this mesh to a point cloud 7 or four multi-view images 8. We claim that the obtained results look promising, and this practical pipeline opens new opportunities of in-the-wild CAD reconstruction.

**Failure cases**    are depicted in Fig. 9. `cadrille` always predicts a geometrically relevant shape, but still might miss details, especially for objects with complex and granular surfaces.

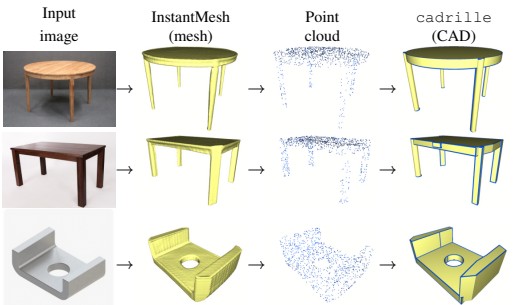

Figure 7: Results of CAD reconstruction from a *single* real-world image. `cadrille` takes point clouds sampled from mesh reconstructed by InstantMesh as input.

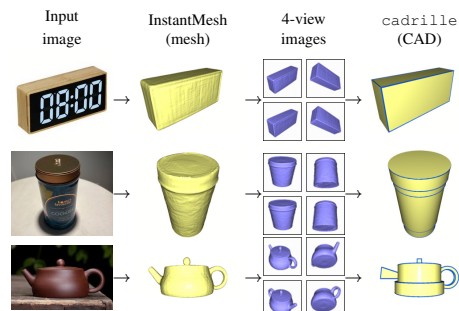

Figure 8: Results of CAD reconstruction from a *single* real-world image. `cadrille` takes multi-view images rendered from mesh reconstructed by InstantMesh as input.

## B    QUANTITATIVE RESULTS

**Results on Omni-CAD**    Omni-CAD is claimed to be the first multimodal dataset featuring point clouds, multi-view images, and textual descriptions of CAD models. However, texts are not fully utilized for CAD reconstruction, since generated models are only assessed in a user study and no standard quantitative metrics are provided (Xu et al., 2024b). Accordingly, we validate `cadrille` only in point cloud-based and image-based scenarios, and report the evaluation results in Tab. 4.

|  | Text | Text2CAD | cadrille | GT |
|--|------|----------|----------|-----|

*Create a new coordinate system with Euler angles set to 0 degrees for the first 2 angles and -90 degrees for the third angle. Set the translation vector to 0.2766, 0.1476, and 0.2766. On the first face, draw a 2-dimensional sketch consisting of 2 loops. For the first loop, draw a circle with its center at 0.0984, 0.0984, and a radius of 0.0984. For the second loop, draw another circle with the same center but a smaller radius of 0.048. Scale the entire 2-dimensional sketch by a factor of 0.1969. Rotate and translate the scaled 2-dimensional sketch using the previously defined coordinate system settings. Extrude the outer loop (the larger circle) by 0.1476 units in the direction opposite to the normal. Similarly, extrude the inner loop (the smaller circle) by the same distance in the same direction. Ensure that the extrusion results in a new solid body. The final dimensions of the cylindrical object with a hole should be a length of 0.1969 units, a width of 0.1969 units, and a height of 0.1476 units. The object resembles a doughnut shape, with a smooth surface and a uniform diameter.*

*Create a new coordinate system with Euler Angles set to [0, 0, 0] and a Translation Vector of [0, 0, 0]. Draw a 2D sketch on the XY plane and define the first face using a loop of 4 lines. The first line starts at (0, 0) and ends at (0.5, 0). The second line starts at (0.5, 0) and ends at (0.5, 0.75). The third line starts at (0.5, 0.75) and ends at (0, 0.75). The fourth line starts at (0, 0.75) and ends at (0, 0). Scale the 2D sketch using a scale factor of 0.75. Transform the scaled 2D sketch into 3D using the same Euler Angles and Translation Vector. Extrude the 2D sketch to create a 3D model with an extrusion depth of 0.5 units towards the normal. Create a new solid body with this extrusion and verify the dimensions: length of 0.5 units, width of 0.75 units, and height of 0.5 units.*
*Next, create a new coordinate system with Euler Angles set to [0, 0, 0] and a Translation Vector of [0.1, 0, 0.5]. Draw a 2D sketch on the XY plane and define the first face using a loop of 4 lines. The first line starts at (0, 0) and ends at (0.3, 0). The second line starts at (0.3, 0) and ends at (0.3, 0.75). The third line starts at (0.3, 0.75) and ends at (0, 0.75). The fourth line starts at (0, 0.75) and ends at (0, 0). Scale the 2D sketch using a scale factor of 0.75. Transform the scaled 2D sketch into 3D using the same Euler Angles and Translation Vector. Extrude the 2D sketch to create a 3D model with an extrusion depth of 0.4 units in the opposite direction of the normal. Remove material from the existing body using this extrusion and verify the dimensions: length of 0.3 units, width of 0.75 units, and height of 0.4 units.*
*The final shape is a U-shaped bracket with a rectangular cross-section. It has 2 parallel sides and an open space in the middle. The dimensions are: length of 0.5 units, width of 0.75 units, and height of 0.5 units.*

*Create a new coordinate system with Euler angles set to 0, 0, and -90 degrees, and a translation vector of 0, 0.2812, and 0.1406. On the first face, draw the first loop as a rectangle with the following lines: the first line starts at (0, 0) and ends at (0.3398, 0); the second line starts at (0.3398, 0) and ends at (0.3398, 0.4687); the third line starts at (0.3398, 0.4687) and ends at (0, 0.4687); the fourth line starts at (0, 0.4687) and ends at (0, 0). In the second loop, draw a circle with a center at (0.2344, 0.2344) and a radius of 0.0937. In the third loop, draw a circle with a center at (0.2344, 0.082) and a radius of 0.0176. In the fourth loop, draw a circle with a center at (0.2344, 0.3867) and a radius of 0.0176. Apply a scale factor of 0.4687 to the entire sketch. Rotate the scaled sketch using the Euler angles set in the coordinate system and translate it using the translation vector. Extrude the sketch 0.2812 units along the normal direction without extruding in the opposite direction to create a new solid body. The final dimensions of the rectangular box with circular cutouts are a length of 0.3398 units, a width of 0.4687 units, and a height of 0.2812 units.*

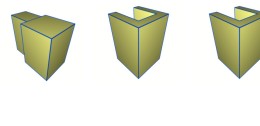

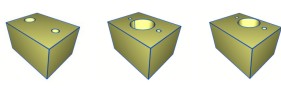

Figure 6: Results of text-based CAD reconstruction on the DeepCAD dataset.

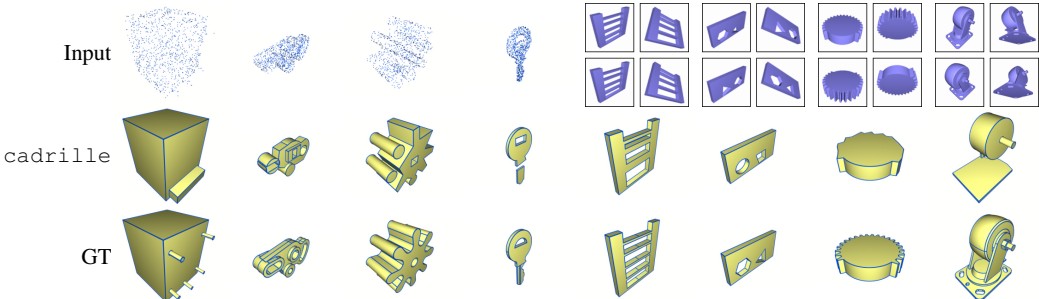

Figure 9: Failure cases of CAD reconstruction from point clouds and multi-view images on Deep-CAD, Fusion360, and CC3D datasets.

| Method | RL | Point Cloud | | | Multi-view Images | | |
|--------|----|-----|------|-----|-----|------|-----|
|  |  | CD↓ | IoU↑ | IR↓ | CD↓ | IoU↑ | IR↓ |
| DeepCAD | ✗ | 45.1 |  | 5.7 |  |  |  |
| CAD-MLLM | ✗ | 18.5 |  | 1.3 | 37.7 |  |  |
| cadrille | ✗ | 1.00 | 79.1 | 3.7 | 1.22 | 77.0 | 6.2 |
| cadrille | ✓ | **0.77** | **84.2** | **0.6** | **0.60** | **84.8** | **0.0** |

Table 4: Results of CAD reconstruction from point clouds and multi-view images from the Omni-CAD dataset. We specify *mean* CD since it is the only CD metric reported by CAD-MLLM.

| Method | Train | CD↓ | IoU↑ | IR↓ |
|--------|-------|-----|------|-----|
| DeepCAD | $D_p$ | 89.2 | 39.9 | 25.2 |
| MultiCAD | $D_p$ | 42.2 |  | 16.5 |
| HNC-CAD | $D_p$ | 36.8 | 63.5 | 7.3 |
| TransCAD | $D_p$ | 33.4 | 60.2 | 2.4 |
| CAD-Diffuser | $D_p$ | 3.85 | 63.2 | 1.7 |
| CAD-SIGNet | $D_p$ | 0.70 | 58.3 | 9.3 |
| **cadrille** | $D_p$ | **0.66** | **63.7** | **0.6** |
| CAD-Recode | $R_p$ | 0.19 | 79.1 | 5.0 |
| **cadrille** | $R_p$ | **0.19** | **79.8** | **2.8** |

Table 5: Results of point-based CAD reconstruction on the Fusion360 test set. All reported metrics are obtained using an SFT model without RL.

The metrics of prior works are cited in accordance to CAD-MLLM; we multiply their values by 10 for consistency, since they report all scores at $10^2$ scale, while we use $10^3$ scale in all tables. Here, CAD-MLLM results are obtained through training on the Omni-CAD dataset, while `cadrille` is evaluated in a pure cross-dataset scenario without any adaptation. In general, the results on Omni-CAD are similar to the ones obtained on other benchmarks, namely, `cadrille` achieves the highest IoU score among all tested approaches, while IR is the lowest with the large margin, reaching 0.0 on image-based benchmark due to RL fine-tuning.

**More baselines on Fusion360** In Tab. 5, we compare `cadrille` to more point cloud-based CAD reconstruction baselines on the Fusion360 dataset. As can be observed, `cadrille` outperforms competitors trained either on DeepCAD or CAD-Recode datasets. The only existing three-modal approach CAD-MLLM reports only mean CD for the Fusion360 dataset, so we list it among other approaches in another Tab. 6. As can be observed, CAD-MLLM far behind our method, with mean CD value being two orders of magnitude larger than of `cadrille`.

**Mean CD** Mean CD is sometimes reported alongside *median* CD. In the main paper, we only provided median CD as the most common CAD reconstruction metric. Nevertheless, the evaluation protocol might be considered incomplete without mean CD. In Tab. 6, we report *mean* CD metric for all scenarios on all three datasets. Due to randomized point sampling, mean CD is not 0 even for perfectly accurate predictions: i.e., evaluation of ground truth CAD models from the DeepCAD dataset gives 0.16, Fusion360 – 0.13 and CC3D – 0.15; as can be observed, while `cadrille` delivers the lowest CD scores, there is room for improvement. Our SFT model outperforms all competitors. The gain is especially tangible in text-based CAD reconstruction, where Text2CAD mean CD is reduced 6x from 26.4 to 3.95. With RL fine-tuning (Dr. CPPO) the new state-of-the-art is set in both image-based and point cloud-based CAD reconstruction on all three datasets. The most dramatic improvement is demonstrated on the Fusion360 dataset, where the relative increase exceeds 40% (Tab. 5).

| Method | Train Data | RL | DeepCAD Points | DeepCAD Images | Text | Fusion360 Points | Fusion360 Images | CC3D Points | CC3D Images |
|---|---|---|---|---|---|---|---|---|---|
| PointNet→DeepCAD | $D_p$ | ✗ | 42.5 | | | 76.1 | | | |
| CAD-MLLM | $O_{pit}$ | ✗ | | | | 33.9 | | | |
| CAD-SIGNet | $D_p$ | ✗ | 6.81 | | | 14.5 | | 32.6 | |
| CAD-Recode | $R_p$ | ✗ | 0.83 | | | 1.21 | | 3.21 | |
| LRM→CAD-Recode | $R_p$ | ✗ | | 3.36 | | | 4.33 | | 4.75 |
| BERT→DeepCAD | $D_t$ | ✗ | | | 97.9 | | | | |
| CAD-Coder | $D_t$ | ✗ | | | 74.6 | | | | |
| Text2CAD | $D_t$ | ✗ | | | 26.4 | | | | |
| Text-to-CadQuery | $D_t$ | ✗ | | | 11.8 | | | | |
| `cadrille` | $D_{pit}$ | ✗ | 3.43 | 3.57 | 4.24 | 7.61 | 8.59 | 12.2 | 13.2 |
| `cadrille` | $R_{pi}+D_t$ | ✗ | **0.76** | **0.81** | **3.95** | **1.10** | **1.13** | **2.32** | **3.50** |
| CADFusion | $D_t$ | DPO | | | 19.9 | | | | |
| CAD-Coder | $D_t$ | GRPO | | | 6.54 | | | | |
| `cadrille` | $R_{pi}$ | DPO | 0.61 | 1.35 | | 0.84 | 1.27 | 2.32 | 3.33 |
| `cadrille` | $R_{pi}$ | Dr. CPPO | **0.57** | **0.43** | | **0.58** | **0.64** | **1.86** | **2.68** |

Table 6: *Mean* CD scores obtained across all benchmarks and available input modalities. RL fine-tuning is performed using $D_i^-+F_i^-$ data.

| Method | CD↓ | IoU↑ | IR↓ |
|---|---|---|---|
| GPT-4o | 62.6 | | 64.4 |
| CAD-GPT | 9.77 | | 1.6 |
| DINOv2→HNC-CAD | 2.14 | | 11.4 |
| Img2CAD | 1.60 | | 28.8 |
| DINOv2→DeepCAD | 1.26 | | 12.3 |
| CADCrafter | 0.72 | | 8.1 |
| cadrille | **0.21** | **81.7** | **1.3** |

Table 7: Results of CAD reconstruction from a *single* image on the DeepCAD dataset. All reported metrics are obtained with an SFT model without RL.

| Method | CD↓ | IoU↑ | IR↓ |
|---|---|---|---|
| GPT-o4-mini | 2.37 | 60.4 | 15.9 |
| cadrille | **0.17** | **92.2** | **0.0** |

Table 8: Results of CAD reconstruction from multi-view images on the Deep-CAD dataset.

**Single-image CAD reconstruction**   To compare against single-image CAD reconstruction methods, namely, CAD-GPT (Wang et al., 2025b) and Img2CAD (Chen et al., 2024), we conduct an experiment with single-view images on the DeepCAD test set. Our SFT model improves over CAD-Crafter (Chen et al., 2025), reducing median CD from 0.72 to 0.21. As could be expected, the results of single-view CAD reconstruction are slightly inferior to the ones obtained from multi-view images (81% vs 86% IoU).

**Zero-shot CAD reconstruction**   Image-based CAD reconstruction can be performed in zero-shot way using existing VLMs. CAD-GPT sets a weak baseline using GPT-4o, that has an invalidity ratio of 64% Tab. 7

We construct another baseline for CAD reconstruction from multi-view images. Four images rendered with orthogonal viewing directions are given to GPT-o4-mini to produce a Python code of CAD model. We apply iterative closest point (ICP) to align predictions before computing metrics so that correct predictions with wrong orientation are not penalized. As can be seen in Tab. 8, this strategy allows achieving significantly better results compared to CAD-GPT. Still, invalidity ratio is as high as 15% and IoU is 30% lower compared to our cadrille.

**Inference time**   When target at practical use, efficiency is an issue. In Tab. 9, we compare inference time of our multimodal cadrille against previous best single-modal methods: CAD-Recode, Text2CAD, and our baseline LRM→CAD-Recode.

Our cadrille is built on top of Qwen2-VL-2B, the smallest Qwen2 model with vision capabilities. When inferred on point clouds, it is 20% slower than CAD-Recode using Qwen2-1.5B. The image inference takes comparable time to proceed. Processing text prompts from Text2CAD dataset lasts notably longer, so that the inference time almost doubles and reaches 3.9 seconds. Text2CAD uses a smaller and faster BERT-large model, that allows achieving efficiency at cost of accuracy. Compared to Text2CAD, cadrille delivers 6x better mean CD 6 being only 2x slower.

## C   ABLATION EXPERIMENTS

**RL fine-tuning vs test-time sampling?**   A natural concern about the RL fine-tuning is that it might compromise the diversity of responses, hence affecting the final prediction quality. To address this, we conduct an ablation study to investigate how the number of test-time samples affects the performance of both CAD-Recode and cadrille. As reported in 11, CAD-Recode produces more accurate results with an increasing number of samples, but remains inferior to cadrille. On all the benchmarks, cadrille with 1 sample consistently outperforms CAD-Recode with 2 samples in both accuracy and invalidity ratio. In terms of IR, cadrille with 1 sample is better than CAD-Recode with as many as 10 samples, which actually proves RL to boost the robustness of the model.

As could be expected, the gap between IoU@1 with and without RL being significantly larger than the gap between IoU@k (e.g. $k = 10$) with and without RL. This result aligns perfectly with the

recent evidence coming from the math domain (see Fig. 2 of Yue et al. (2025) and Fig. 4 of Liu et al. (2025a)).

| Method | LLM | Points | Images | Text |
|---|---|---|---|---|
| CAD-Recode | Qwen2-1.5B | 1.8 | | |
| LRM→CAD-Recode | Qwen2-1.5B | | 4.4 | |
| Text2CAD | BERT-336M | | | 1.7 |
| cadrille | Qwen2-VL-2B | 2.0 | 2.0 | 3.9 |

Table 9: Inference time in seconds measured on the DeepCAD dataset. All methods are benchmarked on the single H100 GPU with a batch size of 1.

| $K$ | DeepCAD | Fusion360 | CC3D |
|---|---|---|---|
| 2 | 77.3 | 70.0 | 49.7 |
| 3 | 86.2 | 77.8 | 55.6 |
| 5 | **86.9** | **78.5** | **56.0** |

Table 10: Results of image-based CAD reconstruction on three datasets, obtained with varying number of samples $K$ in DPO. IoU scores are reported.

| | DeepCAD | | | | | | Fusion360 | | | | | | CC3D | | | | | |
|---|---|---|---|---|---|---|---|---|---|---|---|---|---|---|---|---|---|---|
| # samples | 1 | | 10 | | | | 1 | | 10 | | | | 1 | | 2 | | 10 | |
| | IoU↑ | IR↓ | IoU↑ | IR↓ | IoU↑ | IR↓ | IoU↑ | IR↓ | IoU↑ | IR↓ | IoU↑ | IR↓ | IoU↑ | IR | IoU↑ | IR↓ | IoU↑ | IR↓ |
| CAD-Recode | 87.1 | 3.1 | 89.9 | 0.9 | 92.0 | 0.4 | 79.1 | 5.0 | 83.6 | 1.3 | 87.8 | 0.5 | 60.5 | 9.8 | 64.5 | 1.7 | 74.2 | 0.3 |
| cadrille | **90.2** | **0.0** | **91.6** | **0.0** | **93.1** | **0.0** | **85.0** | **0.2** | **86.8** | **0.1** | **89.1** | **0.0** | **67.9** | **0.2** | **70.9** | **0.2** | **74.7** | **0.1** |

Table 11: Results of point-based CAD reconstruction with test-time sampling.

**Data for RL fine-tuning** As described in Sec. 4.4 in the main paper, not all available data is used for RL fine-tuning, but only the most hard examples are considered. By varying the hard-mining threshold $R_{\text{th}}$, we control the difficulty level and the ratio of data selected for fine-tuning. Finding the proper balance is crucial for RL fine-tuning, since it has a notably larger time- and memory footprint compared to SFT, and is hardly feasible without hard example mining.

In Tab. 12, we report results obtained when fine-tuned on data of different volume. All metrics improve gradually with an increase of the amount of the training data. Still, all results reported in this table supersede the SFT baseline (Tab. 2 of the main paper).

**Number of samples in DPO** We investigate how the number of samples used for RL fine-tuning affects the final results. In Tab. 10, we report IoU scores for image-based reconstruction on all three datasets after 20 epochs of DPO fine-tuning. Performance is unstable with only $K = 2$ samples, while our default value of $K = 5$ yields the best performance. The difference between $K = 3$ and $K = 5$ is less than 1% of IoU, suggesting that the model saturates with few samples, so 5 samples are sufficient.

## D    IMPLEMENTATION DETAILS

**Architecture** Our model is built on top of Qwen2-VL Wang et al. (2024) and uses its native capabilities of image and text understanding. In all experiments on multi-view image CAD reconstruction, we use four images. Images are rendered with fixed camera positions, and concatenated into 2x2 grid, forming a combined image of size $268 \times 268$ px (as shown in Fig. 8 and 9). This combined image is passed through the Qwen vision encoder, that outputs 400 input tokens. The point cloud injecting into LLM is implemented exactly as in CAD-Recode. Specifically, our input consists of 256 unordered 3D points without normals, sampled from the surface using the furthest point sampling method. The points are projected into shared embedding space with a single linear layer.

**Training** SFT in cadrille mostly follows the training procedure of CAD-Recode. The only difference is using batch size of 8 and four gradient accumulation steps due to increase of memory for longer prompts in the multimodal scenario. The SFT model is trained with AdamW optimizer for 120k steps and learning rate of 2e-4 on a single H100 GPU.

**RL fine-tuning** The RL fine-tuning hyperparameters are listed in Tab. 13 (DPO) and 14 (Dr. CPPO). In each experiment, the model is initialized with the weights of an SFT model trained

| $R_{\text{th}}$ | RL Data | | DeepCAD | | | Fusion360 | | | CC3D | | |
|---|---|---|---|---|---|---|---|---|---|---|---|
| | DeepCAD | Fusion360 | CD↓ | IoU↑ | IR↓ | CD↓ | IoU↑ | IR↓ | CD↓ | IoU↑ | IR↓ |
| 1 | 10k | 0.7k | 0.18 | 89.0 | 1.0 | 0.18 | 81.0 | 2.6 | 0.74 | 59.8 | 6.3 |
| 3 | 20k | 1.4k | 0.17 | 90.4 | 0.7 | 0.18 | 82.3 | 0.9 | 0.63 | 62.4 | 4.0 |
| 5 | 30k | 2k | 0.17 | 90.9 | 0.2 | 0.17 | 83.5 | 0.5 | 0.62 | 62.3 | 0.9 |
| 6.2 | 40k | 2.5k | 0.17 | 91.2 | 0.1 | 0.17 | 83.9 | 0.2 | 0.60 | 63.3 | 0.7 |
| 7.5 | 50k | 3k | **0.17** | **92.2** | **0.0** | **0.17** | **84.6** | **0.0** | **0.57** | **65.0** | **0.1** |

Table 12: Results of CAD reconstruction from multi-view images with varying amount of data used for RL fine-tuning.

on point clouds and images, and then fine-tuned only on images. All fine-tuning experiments are performed on 8 H100 GPUs.

| Hyperparameter | Value |
|---|---|
| Optimizer | Adam |
| Number of epochs | 20 |
| Batch size | 160 |
| Learning rate | 1e-05 |
| KL regularization coef ($\beta$) | 0.3 |
| Number of samles ($K$) | 5 |

Table 13: DPO tuning hyperparameters.

| Hyperparameter | Value |
|---|---|
| Optimizer | Adam |
| Number of epochs | 20 |
| Batch size | 128 |
| Learning rate | 3e-05 |
| Updates per batch | 3 |
| PPO $\epsilon$ | 0.1 |
| GRPO group size ($G$) | 16 |
| CPPO number of samples ($N$) | 4 |

Table 14: RL fine-tuning hyperparameters.

## E   MISCELLANEOUS

**Python codes**   In Fig. 10, we provide Python code with CadQuery library, produced by `cadrille`. When executed, these Python scripts generate CAD models. Evidently, `cadrille` is not limited to basic geometric primitives from the DeepCAD dataset (such as line, arc, circle), but is also capable of producing more advanced shapes from the CAD-Recode dataset (box, rectangle, cylinder).

**CC3D scans**   Following CAD-Recode, we treat experiments with CC3D dataset as an emulation of a real-world experiment. CC3D contains scans of the physical objects paired with ground truth CAD models. As shown in Fig. 11, CC3D scans contain artifacts such as surface noise, smoothed edges, and missing parts. In our experiments, we sample points from the surfaces of these real noisy scans instead of the perfect CAD models, which essentially makes the experimental scenario much more realistic. Besides, CC3D contains generally more complex CAD models, which are constructed using operations beyond simple extrusion, including revolution, chamfer, and fillet.

```python
import cadquery as cq
w0=cq.Workplane('XY',origin=(0,0,11))
r=w0.sketch().segment((-100,23),(-43,-23)).segment((-63,-95)).segment((0,-55))
    .segment((62,-96)).segment((42,-23)).segment((100,23)).segment((27,26))
    .segment((0,96)).segment((-26,26)).close().assemble().finalize().extrude(-23)
```

```python
import cadquery as cq
w0=cq.Workplane('XY',origin=(0,0,88))
r=w0.sketch().push([[(-87,-89)]]).rect(20,22).push([[(-87,89)]]).rect(20,22)
    .push([[(88,-89)]]).rect(20,22).push([[(88,90)]])
    .rect(20,20).finalize().extrude(-176)
  .union(w0.workplane(offset=-15/2).box(200,200,15))
```

```python
import cadquery as cq
w0=cq.Workplane('YZ',origin=(-63,0,0))
r=w0.sketch().circle(33).circle(27,mode='s').finalize().extrude(-37)
    .union(w0.sketch().segment((-55,-67),(55,-67)).segment((55,-66)).arc((87,0),
    (55,66)).segment((-9,67)).segment((-55,67)).arc((-87,1),(-55,-67)).assemble()
    .push([[(-53,-39)]]).circle(12,mode='s').push([[(-53,39)]]).circle(11,mode='s')
    .push([[(53,-43)]]).circle(12,mode='s').finalize().extrude(21))
  .union(w0.sketch().circle(34).circle(27,mode='s').finalize().extrude(163))
```

```python
import cadquery as cq
w0=cq.Workplane('XY',origin=(0,0,56))
w1=cq.Workplane('XY',origin=(0,0,26))
r=w0.workplane(offset=-149/2).cylinder(149,18.5)
    .union(w0.sketch().circle(100).circle(78,mode='s').finalize().extrude(-111))
    .union(w1.workplane(offset=-52/2).cylinder(52,77))
    .union(w1.workplane(offset=6/2).moveTo(97,0).box(3.5,9.5,6))
```

Figure 10: `cadrille` predictions on DeepCAD, Fusion360, and CC3D datasets. Each row contains predicted CadQuery Python code and its result after execution in Python interpreter.

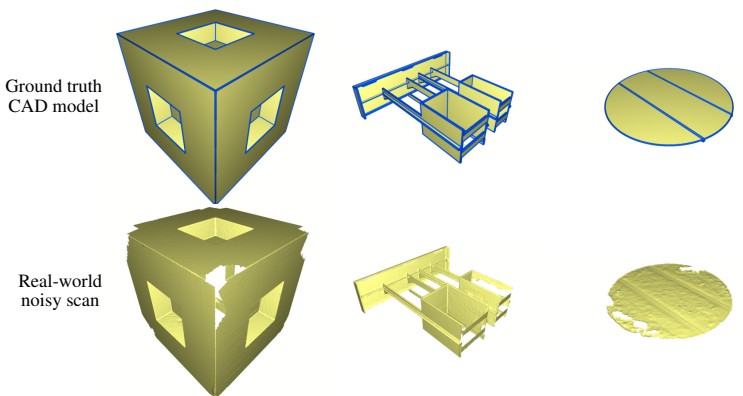

Figure 11: Ground truth CAD models (top row) and noisy scans (bottom row) from the real-world CC3D dataset.

