# OpenReview forum: "cadrille: Multi-modal CAD Reconstruction with Reinforcement Learning"
_ICLR.cc/2026/Conference — ICLR 2026 Conference Desk Rejected Submission_

### Official Review · Reviewer_aZn7 · 2025-10-20

**Soundness:** 4
**Presentation:** 3
**Contribution:** 3
**Rating:** 8
**Confidence:** 4

**Summary:**

This paper presents a multi-modal CAD reconstruction model that takes point clouds, images and text as input modalities and outputs executable python code in the CADQuery language.

A pre-trained Qwen2-VL-2B is first fine tuned on synthetic data and then reinforcement learning (RL) fine tuning employed, using datasets of CAD models create by humans.  This final RL fine-tuning stage does not require a CAD command sequence and could even be performed using mesh data.  This rather unique capability has the potential to greatly expand the amount of data available for training.

The method is evaluated on four datasets, including one containing realistic point clouds.  The model is shown to be state of the art when compared with 10 benchmark methods.

**Strengths:**

This work sets new state-of-the-art for the generation of CAD models using the "CAD command sequence" approach.

The large number of benchmark methods and datasets used for evaluation is impressive.

The ability to train using CAD data from step files or even just 3D meshes is a big benefit.

The good performance on the realistic CC3D is particularly encouraging

It's also interesting and exciting  that  hard example mining was found to give faster convergence during RL fine-tuning

**Weaknesses:**

Figures 1 and 2 are not very inspiring.  While they include all the important details, they somehow don't make the visual impact which should be possible given the quality of the results.  Perhaps just adding more images to them would help.

The quantitative comparison between DPO and Dr. CPPO were welcome, but leave me wondering about the decision to use Dr. CPPO over vanilla Dr GRPO or CPPO.  Clearly online RL was beneficial here, but it's hard to tell if the Dr. CPPO algorithm has any benefit over more traditional methods.  I guess that vanilla GRPO wouldn't be possible because of the need to keep the reference model in vram for the KL loss.

**Questions:**

Dr. CPPO
The expensive part of a CAD RL workflow is usually sampling from the model, rebuilding the CAD and computing the rewards.  Was there any evidence that dropping all but the N samples with the most extreme advantages helped?   More details would be very interesting.

It's a big advantage of this method that the CAD command sequence is not required for the RL fine tuning phase.   This would open up the possibility of training on the large ABC dataset.  I would be very interested to hear about results for this kind of experiment.  Are there problems with unsupported features in the CADQuery language  like edge blends?

Abstract:
"...in the DeepCAD benchmark...
I suggest "... on the DeepCAD benchmark"

Line 211
"...the invalidity ratio (IR) is to 10%, "
Should this be “is up to 10%”?

---

> ### Author Response · Authors · 2025-11-17
>
> We thank the reviewer for the thoughtful assessment and for highlighting the strengths of our method, especially the value of RL without CAD sequences and the strong performance on CC3D. We address the technical questions below; typos will be fixed in the updated manuscript.
>
> ---
> **More details on contributions of Dr. GRPO and CPPO on top of vanilla GRPO.**
>
> As the reviewer correctly notes, both modifications are chosen primarily for efficiency (L. 285):
>  - Dr. GRPO removes the reference model, significantly reducing VRAM usage;
>  - CPPO discards all but the top-k samples, reducing the number of expensive forward passes.
>
> Given that CAD RL is dominated by sampling + CAD compilation time, our full Dr. CPPO pipeline already trains for ~7 days on our hardware (L. 971). Therefore, we did not run vanilla GRPO nor isolate accuracy differences – our focus was achieving feasible training time under strict compute constraints.
>
> ---
> **Can the RL stage be processed on a larger ABC dataset? Will it raise problems with unsupported CAD features?**
>
> We fully agree this is a promising direction and explicitly list it as future work (L. 485). Our current experiments already validate the key idea: RL fine-tuning successfully consumes datasets without CAD programs, namely DeepCAD and Fusion360 (Tab. 12). Fusion360, in particular, contains challenging operations – ellipses, splines, taper extrusions – that do not appear in CAD-Recode at all. RL still approximates these shapes using the available CAD-Recode operators, indicating that our reward-driven policy can robustly fit unseen geometry despite missing high-level operations. Scaling to ABC should therefore be feasible in principle, with the caveat that complex fillets, blends, or freeform operations may require more expressive CAD primitives or longer programs.

---

### Official Review · Reviewer_SjgS · 2025-10-24

**Soundness:** 1
**Presentation:** 3
**Contribution:** 2
**Rating:** 2
**Confidence:** 4

**Summary:**

This paper introduces a multimodal CAD reconstruction model that unifies point clouds, multi-view images, and textual descriptions as input modalities to generate executable Python code for parametric CAD models.

Building upon a multi-modal architecture, cadrille adopts a two-stage training pipeline: supervised fine-tuning (SFT) on large-scale procedurally generated CAD data followed by reinforcement learning (RL) fine-tuning using handcrafted data.

Evaluated on four datasets (DeepCAD, Fusion360, CC3D, and Omni-CAD), cadrille achieves state-of-the-art results in 10 benchmarks spanning all three input modalities.

**Strengths:**

1. Writing and presentation of this paper are highly complete and easy to follow.

2. Comparison and ablation of this paper are thorough and comprehensive.

3. Datasets involved in experiments are various.

**Weaknesses:**

**The reviewer has several serious concerns on some statements in this paper the reliability of evaluation and experiments:**

1. Statement at Lines 52-53:

> "However, the first multimodal methods in that vein (Xu et al., 2024b; Wang et al., 2025b) are dramatically inferior to single-modal approaches, so the full potential of VLMs for CAD reconstruction is yet to be unleashed."

And statement at Line 139:

> "Recent CAD-GPT (Wang et al., 2025b) predicts a CAD model given a single image and textual description, while CAD-MLLM (Xu et al., 2024b) pioneers three-modal CAD reconstruc- tion, yet both these methods fall behind single-modal state-of-the-art results (Rukhovich et al., 2024; Khan et al., 2024b; Chen et al., 2025) by a large margin (up to two orders of magnitude!)."

Might be improper. The authors claim that single-modal state-of-the-art methods' performance is "up to two orders of magnitude!". However, since CAD-GPT and CAD-MLLM have not yet been publicly released as far as I know, it is unclear how this quantitative comparison was obtained. **Without transparent and reproducible evaluation settings aligned across all methods, such a strong claim (especially “up to two orders of magnitude!”) might be misleading.**

Moreover, geometric evaluations—such as those based on Chamfer Distance—are typically preceded by a scaling operation that normalizes shapes into a common coordinate space (e.g., $[−1, 1]^3$ or $[-100, 100]^3$). Differences in normalization schemes, evaluation implementations, or testing environments can significantly affect the absolute magnitude of reported errors.

For example, comparing error values across differently normalized spaces (e.g., $[−100, 100]^3$ vs. $[−1, 1]^3$) is not meaningful without proper alignment or clarification. **Authors are generally expected to maintain consistent scaling in the normalized space across all compared methods to ensure a fair and meaningful evaluation.** (But the reviewer doesn't find the elaboration on the scale of this normalized coordinate space used in this paper's experiment.)

2. Building on the previous concern, the reviewer only observes that in Line 322:

> "CD values are multiplied by $10^3$"

**The authors have not yet specified the scale of the normalized coordinate space used for Chamfer Distance (CD) evaluations in the main text.**

Line 809 says that “we report mean CD since it is the only CD metric reported by CAD-MLLM”.

CAD-MLLM reports its normalized space is $[-0.5, 0.5]^3$. In Table 4, is the scale of normalized space identical to $[-0.5, 0.5]^3$ for all tested methods? (including DeepCAD, CAD-MLLM, Point2CAD, cadrille, any other baselines or methods)

The reviewer is therefore concerned that the comparisons of CD values across methods may not be valid **unless all approaches were evaluated in the same normalized space**. The reviewer thinks it's also essential to explain why LLM-based method cadrille outperforms Point2CAD with one orders of magnitude, since Point2CAD has a setting to fitting/reconstruct input point cloud with paramatric geometry primitive. (Since the proposed method is reported to be better than Point2CAD, how is comparison with sota-method [1] within that setting?)

Consistent normalization is essential for meaningful quantitative comparison, and this detail should be explicitly clarified.

Although the experiments are comprehensive, **the reviewer has serious concerns about the validity of the reported results due to the issues outlined above**—particularly the lack of clarity regarding normalization protocols and the unverifiable performance claims against unreleased methods.

References:

[1] Split-and-Fit: Learning B-Reps via Structure-Aware Voronoi Partitioning, Liu et al. (https://arxiv.org/abs/2406.05261)

**Questions:**

1. Between lines 393 and 395, the results appear identical across all columns except for the IR of CAD-Recode when conditioned on the point cloud. Is this result typed correctly?

(The reviewer gives a score of 2 because of **serious concerns about the correctness and reliability of the paper’s experiments**. If these issues are clearly explained and properly addressed, the reviewer is open to raising the score. The reviewer acknowledges the authors’ efforts in presenting comprehensive experiments and analysis. However, for the conclusions to be meaningful and useful to the research community, the evaluation must be methodologically sound and correct. Once the reviewers' concerns on experiments are resolved, the reviewer may also have additional questions about the paper’s core contribution.)

---

> ### Author Response · Authors · 2025-11-17
>
> We thank the reviewer for the careful reading of our paper and for acknowledging the completeness of our experiments and presentation. We address the concerns below and clarify the evaluation protocol in detail. All evaluation code is provided in the Supplementary material and can be directly inspected.
>
> ---
> **Normalization procedure for CD evaluation is not specified.**
>
> We follow the evaluation protocol from DeepCAD. Exactly the same protocol is used in CAD-Recode [1], Text2CAD [2], CAD-SIGNet [3] – since their source codes are published, this fact can be checked trivially. We attach our code in Supplementary so that Reviewers could try it out and verify the coherence of calculations.
>
> The standard evaluation procedure implies that:
>  - ground truth meshes are rescaled into the unit cube $\[-0.5, 0.5\]^3$,
>  - CD values are calculated for this unit cube scale and multiplied to $10^3$ (L. 322).
>
> In the paper, we do not describe the normalization for brevity; it seems redundant, since we simply follow the common practice and do not change any part of it. Actually, normalization details are also not mentioned in [1, 2, 3], referring the interested reader to the code.
>
> ---
> **Why is the proposed method 2 orders of magnitude better compared to CAD-MLLM?**
>
> The “two orders of magnitude” gap does not originate from our work – it already appears in the literature:
>  - CAD-MLLM performs on par with DeepCAD on the Fusion360 dataset;
>  - CAD-Recode achieves CD values ~100х lower than DeepCAD on Fusion360 $\Rightarrow$ we can assume a comparable difference between CAD-Recode and and CAD-MLLM;
>  - our SFT $\texttt{cadrille}$ exactly matches CAD-Recode $\Rightarrow$ we also expect to obtain ~100x lower CD than of CAD-MLLM.
>
> We provide comparison with CAD-MLLM on Fusion360 and Omni-CAD benchmarks in Sec. B (Tab. 4 and 6).
>
> ---
> **CAD-MLLM code is not released, so comparison to this work might be misleading.**
>
> CAD-MLLM clearly uses DeepCAD code, since they report scores of the DeepCAD model on their newly presented Omni-CAD dataset. The DeepCAD code contains the evaluation kit, which matches our evaluation procedure. So we can directly compare our scores with the ones reported in CAD-MLLM.
>
> ---
> **Comparison with B-Rep generation methods Point2CAD and Split-and-Fit.**
>
> We clearly position our method in the CAD sequence-prediction family (L. 114). Same as our predecessors [1, 2, 3], we do not compare our method with B-Rep generation approaches (including Point2CAD and Split-and-Fit) in the main paper (Sec. B; L.811).
>
> In the Appendix, we provide comparison with CAD-MLLM on their novel benchmark, Omni-CAD. To establish a baseline, the authors of CAD-MLLM run Point2CAD on Omni-CAD and report the results achieved; we cite this comparison according to their paper. As mentioned by the Reviewer, this could be misleading for methodological consistency, so we will remove this in the revised version of the Appendix.
>
> ---
> **Are the results in L. 393–395 typed correctly?**
>
> Yes. As noted in L. 350-352, adding extra input modalities does not degrade performance, so CD and IoU remain identical when moving from CAD-Recode to $\texttt{cadrille}$. The small difference in IR likely reflects model robustness differences (Qwen2-1.5B vs. Qwen2-VL-2B).
>
> ---
> References
>
> [1] CAD-Recode: Reverse Engineering CAD Code from Point Clouds, ICCV 2025
>
> [2] Text2CAD: Generating Sequential CAD Designs from Beginner-to-Expert Level Text Prompts, NeurIPS 2024
>
> [3] CAD-SIGNet: CAD Language Inference from Point Clouds using Layer-wise Sketch Instance Guided Attention, CVPR 2024

---

> > ### Comment · Reviewer_SjgS · 2025-11-19
> >
> > The reviewer acknowledges the author's reply. Here is my feedback on each section from the author's reply.
> >
> > ---
> >
> > Author's reply:
> >
> > > In the paper, we do not describe the normalization for brevity; it seems redundant, since we simply follow the common practice and do not change any part of it. Actually, normalization details are also not mentioned in [1, 2, 3], referring the interested reader to the code.
> >
> > I understand the reviewer’s concern regarding readability. However, I believe that including brief descriptions of “normalized space” and “sample point” does not reduce clarity. Rather, these additions help make the definitions more precise and improve the overall comprehensibility of the paper. The reader won't be confused by this clarification.
> >
> > ---
> >
> > Author's reply:
> >
> > > The “two orders of magnitude” gap does not originate from our work – it already appears in the literature
> >
> > In the original paper, the claim is **"Recent CAD-GPT (Wang et al., 2025b) predicts a CAD model given a single image and textual description, while CAD-MLLM (Xu et al., 2024b) pioneers three-modal CAD reconstruction, yet both these methods fall behind single-modal state-of-the-art results (Rukhovich et al., 2024; Khan et al., 2024b; Chen et al., 2025) by a large margin (up to two orders of magnitude!)."**
> >
> > In short, this paper claims that **multi-modal CAD reconstruction falls behind single-modal state-of-the-art results(up to two orders of magnitude!)**. So the reviewer wants to see how to prove this strong claim.
> >
> > For each explanation provided by the author,
> >
> > > CAD-MLLM performs on par with DeepCAD on the Fusion360 dataset;
> >
> > This is not correct. CAD-MLLM only claims to train with data from Omni-CAD. And Fusion360 is only used for their generalization test.
> >
> > > CAD-Recode achieves CD values ~100х lower than DeepCAD on Fusion360 => we can assume a comparable difference between CAD-Recode and CAD-MLLM;
> >
> > Why CAD-Recode achieves CD values ~100х lower than DeepCAD on Fusion360 can lead to a strong assumption that CAD-Recode achieves CD values ~100х lower than CAD-MLLM?
> >
> > > Our SFT cadrille exactly matches CAD-Recode. We also expect to obtain ~100x lower CD than of CAD-MLLM.
> >
> > I don't have questions on "Our SFT cadrille exactly matches CAD-Recode". But still, why "We also expect to obtain ~100x lower CD than of CAD-MLLM."?
> >
> > > We provide comparison with CAD-MLLM on Fusion360 and Omni-CAD benchmarks in Sec. B (Tab. 4 and 6).
> >
> > CAD-MLLM hasn't been trained with Fusion360 data from their paper's experiment, but CAD-MLLM does training with Omni-CAD benchmarks.
> >
> > But again, the reviewer highly values the preciseness of the experiment process. Tab. 6's comparison with CAD-MLLM might not be proper here. Tab. 4 might be good, but the reviewer is still concerned about the potential misalignment because CAD-MLLM just doesn't provide code details for their CD computation or potential pre-process or post-process.
> >
> > **Another question**: Since both the DeepCAD and OmniCAD datasets are originally represented in command-sequence format, is the ground-truth model generated using CadQuery exactly identical to the model produced by the original command sequence? (Some cadquery dataset construction pipeline can't achieve the alignment of gt model to my knowledge, like the pipeline in Text-to-CadQuery.)
> >
> > ---
> >
> > > CAD-MLLM clearly uses DeepCAD code, since they report scores of the DeepCAD model on their newly presented Omni-CAD dataset. The DeepCAD code contains the evaluation kit, which matches our evaluation procedure. So we can directly compare our scores with the ones reported in CAD-MLLM.
> >
> > Actually, the reviewer also used to ask for CAD-MLLM CD computation details. And their authors indicate that they make lots of code adaptations to strictly align their CD computation with other B-Rep generation methods, because every single baseline outputs models with misaligned normalized space in their experiment. So the reviewer thinks it might not so much proper to assume/imagine their evaluation code.
> >
> > ---
> >
> > Author's reply:
> >
> > > As mentioned by the Reviewer, this could be misleading for methodological consistency, so we will remove this in the revised version of the Appendix.
> >
> > The reviewer thinks that's fine.
> >
> > ---
> >
> > Author's reply:
> >
> > > Yes. As noted in L. 350-352, adding extra input modalities does not degrade performance, so CD and IoU remain identical when moving from CAD-Recode to cadrille.
> >
> > Thanks for your clarification. Sounds good to me.
> >
> > ---
> >
> > **Some additional questions:**
> >
> > 1. The reviewer trys to seek the proof of the claim of **multi-modal CAD reconstruction falls behind single-modal state-of-the-art results(up to two orders of magnitude!)**. The reviewer's concerns about the comparison with unreleased methods are stated above. The reviewer is curious whether this can be proved by doing an ablation under cadrille codebase with some adaptations to certify this two orders of magnitude drawback for the multi-modal CAD reconstruction method.

---

> > > ### Comment · Reviewer_SjgS · 2025-11-19
> > >
> > > 2. The reviewer acknowledges that certain misalignment issues in Chamfer Distance (CD) computation are difficult to avoid. Could the authors provide the CD between the ground-truth model and itself in Table 1? Because of sampling randomness and varying point counts, the CD between two identical models is not strictly zero. Reporting this “best achievable CD” would give readers a sense of the remaining improvement margin and also help other future researchers verify whether their experimental setup is consistent with Cadrille.

---

> > > > ### Comment · Reviewer_SjgS · 2025-11-19
> > > >
> > > > I raise my rate and confidence because my previous largest concern about the normalized space has been solved. But I still suggest the reviewer to think carefully about how to properly prove the claim mentioned in my additional question 1.

---

> ### Author Response · Authors · 2025-11-22
>
> We are happy that the Reviewer raised their initial score, acknowledging our comments on GT normalization and B-Rep generation methods. Still, we believe that poor results of CAD-MLLM compared to our $\texttt{cadrille}$ is rather evidence of the strength of our method rather than a reason for rejection of our paper. Below, we try to elaborate on the CD calculation once again.
>
> **Actually, the reviewer also used to ask for CAD-MLLM CD computation details. … So the reviewer thinks it might not so much proper to assume/imagine their evaluation code.**
>
> CAD-MLLM CD computation details are provided in the CAD-MLLM paper.
>
> | | CAD-MLLM | $\texttt{cadrille}$ |
> |:-:|:-:|:-:|
> | Normalization | $\[-0.5, 0.5\]^3$ in their Sec. 6.1.4 | $\[-0.5, 0.5\]^3$ in our code |
> | Multiplication | x100 in their Tab. 6 | x1000 in L. 321 |
>
> Taking these details into account, we claim that after additional x10 multiplication (they report x100, we report the same values x1000), their results can be compared to ours.
>
> **This is not correct: CAD-MLLM performs on par with DeepCAD on the Fusion360 dataset.**
>
> Indeed. The matching names of DeepCAD method and DeepCAD dataset might be a source of misunderstanding here; if so, we apologize for vague formulations. We meant that when *tested* on Fusion360, CAD-MLLM scores are within the same order of magnitude as scores of the DeepCAD method, regardless of the training dataset. We consider the CAD-MLLM method as a combination of model architecture + training recipe, which includes the dataset used for training; in the end, we only compare numbers to numbers.
>
> In Tab. 6, we show that the training dataset actually matters. Below, we report results taken from Tab. 6; we do not provide any novel scores but only list the numbers that are already in the paper.
>
> When training on the limited handcrafted DeepCAD dataset, $\texttt{cadrille}$'s CD is only one order of magnitude lower then the one of DeepCAD. Using procedurally generated CAD-Recode dataset for training results in a two orders of magnitude improvement. The same effect is observed by the authors of CAD-Recode is reported in Tab. 3 of the CAD-Recode paper. Furthermore, $\texttt{cadrille}$'s CD score is improved by 2x by reinforcement learning. As a result, the difference with CAD-MLLM is two orders of magnitude.
>
> | Method | SFT data | RL data | mean CD |
> |:-:|:-:|:-:|:-:|
> | DeepCAD | DeepCAD | – |76.1 |
> | CAD-MLLM | Omni-CAD | – | 33.9 |
> | CAD-Recode | CAD-Recode | – | 1.21 |
> | $\texttt{cadrille}$ | DeepCAD | – | 7.61 |
> | $\texttt{cadrille}$ | CAD-Recode | – | 1.10 |
> | $\texttt{cadrille}$ | CAD-Recode | DeepCAD + Fusion360 | **0.58** |
>
> **Are CAD command sequences in DeepCAD and CadQuery formats the same?**
>
> No, as we say in L. 238. However, this is not a limitation, since DeepCAD sequences can be easily converted into CadQuery format, and vice versa. When training $\texttt{cadrille}$ on DeepCAD dataset, we convert it to the CadQuery format (Tab. 1, 2, 3), when training on CAD-Recode, we use the original data that is already stored in the CadQuery format.
>
> **Could the authors provide the CD between the ground-truth CAD model and itself?**
>
> Sure. This is an important upper-bound value, and we will add it to the updated manuscript. Below, we report the mean CD averaged across all test samples in each dataset. $\texttt{cadrille}$ values are taken from Tab. 6. Obviously, there is room for improvement.
>
> | | DeepCAD | Fusion360 | CC3D |
> |:-:|:-:|:-:|:-:|
> | $\texttt{cadrille}$ | 0.57 | 0.58 | 1.86 |
> | upper-bound | 0.16 | 0.13 | 0.15 |

---

> ### Comment · Reviewer_SjgS · 2025-11-22
>
> (Following structure keeps the same as authors' last reply)
>
> **Actually, the reviewer also used to ask for CAD-MLLM CD computation details. … So the reviewer thinks it might not so much proper to assume/imagine their evaluation code.**
>
> Thanks for your table. That's clear to me.
>
> **This is not correct: CAD-MLLM performs on par with DeepCAD on the Fusion360 dataset.**
>
> Thanks for your recall on your table and the highlight of the data used in your experiment. The reviewer realizes the author's intention here. And the last three rows' ablation clearly proves the effectiveness of the proposed RL scheme. But the reviewer still suggest the authors to slightly rewrite "Related work" in L139, because the reviewer is still not in agreement that the **two orders of magnitude** in Line139. To be rigorous, let's recall on original text:
>
> > yet both these methods fall behind single-modal state-of-the-art results (Rukhovich et al., 2024; Khan et al., 2024b; Chen et al., 2025) by a large margin (up to two orders of magnitude!).
>
> Suppose that sota single-modal result here refers to CAD-Recode, then the comparison shall be between CAD-MLLM(33.9) and CAD-Recode(1.21). So this comparison shall be one order of magnitude. (Here should be the survey of old methods.)
>
> And then comes the proposed method(cadrille) successfully enhances the multi-modal method over the single-modal method. To correctly highlight your work's contribution is to make "multi-modal method over the single-modal".
>
> And finally, please consider adding a footnote for CAD-MLLM results on Fusion360, since they are not trained on Fusion360, just inference for a generalization test. (To reflect the actual test setup.)
>
> **Are CAD command sequences in DeepCAD and CadQuery formats the same?**
>
> I know formats are different, but this is not my original question. My original question is the clarification on
>
> > "is the ground-truth model generated using CadQuery exactly identical to the model produced by the original command sequence?"
>
> **Could the authors provide the CD between the ground-truth CAD model and itself?**
>
> That's perfect for me. This upper-bound mainly resolves my previous concern.
>
> ---
>
> And probably **one last thing** that I want to ask, which I think needs elaboration:
>
> For table1, the CD is reported for various dataset. That's good. But that's very close to the upper-bound(mean CD) you provided.
>
> Actually, my previous empirical knowledge for the upper-bound(mean CD) is also about 0.18-0.20. And this initially made it hard to believe that your mean CD in table1 is so close or even better than the upper-bound mean CD.
>
> But for now, the reviewer thinks the authors might miss to elaborate some information in the main body, and that's leading to my previous largest doubts. For L825-826 in the appendix, I just noticed that this paper main body reports the median CD.
>
> > Mean CD is sometimes reported alongside median CD. In the main paper, we only provided median CD as the most common CAD reconstruction metric.
>
> But let get back to the main body, **I don't find elaboration of CD(whether mean or median) from table1's caption or L342-354**. And between L321-322, it writes:
>
> > we report both mean and a more robust median CD, both computed using 8192 point
>
> The missing information in the main body makes me suppose the CD in table1 is mean CD. And the result of value between 0.18-0.20 is so close to my understanding of upper-bound mean CD. That's why I'm confused by the result in Table1. **And this is one of the important things that the authors need to elaborate in the main body, in case there's confusion to the reader.**
>
> ---
>
> Lastly, another thing to mention about your reply:
>
> > Still, we believe that poor results of CAD-MLLM compared to our cadrille is rather evidence of the strength of our method rather than a reason for rejection of our paper.
>
> I **NEVER** threatened to reject your paper. Everything I want to do is to eliminate the thing that I'm concerned about. My previous largest concern mainly comes from "too good mean CD", and I think the confusion of mean CD and median CD is not mainly my problem because it's not elaborated so clearly in the main body or table itself.
>
> Actually I'm quite active and trying to treat every paper as responsibly as possible. If you solve my concern and make this paper better, I will raise my rate. I'm not an inactive reviewer who sticks to the original rating. But I will not contribute my time and volunteer for the review process if I feel further being maliciously speculated about.

---

> ### Author Response · Authors · 2025-11-22
>
> First, we would like to express our sincere gratitude to the Reviewer for their deep dive into our paper, and for active participation in the discussion (during the weekend!).
>
> ---
> **The reviewer still suggests the authors to slightly rewrite “Related work”... two orders of magnitude in L. 139.**
>
> We agree that this claim may be too bold, so we will remove the exact formulation from the paper.
>
> ---
> **Please consider adding a footnote for CAD-MLLM results on Fusion360, since they are not trained on Fusion360**
>
> CAD-MLLM results on Fusion360 are only given in Tab. 6. In that table, we explicitly specify the training dataset in the column *Train Data*, so it can be observed that CAD-MLLM is trained on Omni-CAD ($O_\text{pit}$ in L. 836), not Fusion360. An additional clarification in textual form might be needed, so we will mention this in Sec B. (L. 821).
>
> ---
> **I just noticed that this paper main body reports the median CD**
>
> We admit that it is a source of possible confusion. We specify the metric used only in Sec. B (L. 825): “*In the main paper, we only provided median CD*” but never mention that in the main paper itself. We thank the Reviewer for careful reading of our paper and will edit the manuscript accordingly.
>
> ---
> **Does DeepCAD to CadQuery conversion keep identical CAD model? Alignment is reported to be not perfect in Text-to-CadQuery**
>
> This is a valid concern. Yet, we only need this conversion for an ablation study. Our major contribution is a training scheme that implies using procedurally generated CAD models for SFT and handcrafted meshes for RL; none of these stages relies on handcrafted CAD codes, as comprise the DeepCAD dataset.
>
> Actually, we cannot guarantee that the DeepCAD and CadQuery representations are 100% identical. All operations: 2D (line, arc, and circle), 3D (extrusion), and boolean (union, difference, intersection) are the same, the only source of potential difference is the rounding procedure. While DeepCAD rounds float values to -128…128 integer range, CAD-Recode uses -100…+100 range (Tab. 3, row 2 in the CAD-Recode paper). In $\texttt{cadrille}$, we follow the conversion procedure as in CAD-Recode.
>
> Regarding the conversion procedure in Text-to-CadQuery. We find it extremely questionable, e.g., they introduce a multi-turn procedure with multiple calls to a proprietary Gemini model to simply map a DeepCAD JSON to CadQuery. We achieve the same with a Python script having <100 lines of code, that simply maps *line*, *arc*, *circle* to *moveTo*, *lineTo*, *threePointArc*, *circle*, etc.
>
> ---
> Overall, so far we are willing to update our manuscript as follows:
>  - normalization procedure to $\[-0.5, 0.5\]^3$ in Sec. 5 (L. 322),
>  - remove B-Rep generation method comparison in Sec. B (Tab. 4, L. 803)
>  - report the upper-bound CD for each dataset in Sec. B (L. 825)
>  - specify, that in main paper we report only median CD (L. 322)
>  - remove *up to two orders of magnitude!* from Sec. 2
>  - mention that CAD-MLLM is trained on Omni-CAD in L. 821
>
> If any other comments, concerns or proposals would follow, we are ready to discuss and update the paper accordingly.

---

> ### Comment · Reviewer_SjgS · 2025-11-23
>
> The reviewer appreciates the authors' reply. (Following structure keeps the same as authors' last reply)
>
> **We agree that this claim may be too bold, so we will remove the exact formulation from the paper.**
>
> Pointing out "one order of magnitude" for the comparison between the previous multi-modal and single-modal methods is fine, as I previously mentioned. But that's also fine if the authors want to remove in order to leave some leeway.
>
> ---
>
> **In that table, we explicitly specify the training dataset in the column Train Data, ... An additional clarification in textual form might be needed, ...**
>
> Sound good. If some other methods have a similar experimental setup, this additional clarification deserves to be highlighted, as the authors suggest.
>
> ---
>
> **We admit that it is a source of possible confusion....**
>
> This confusion is the main source of my previous doubt. So it might be better to directly reflect in table1 itself about "median", or highlight with bold text in the caption.
>
> ---
>
> **Our major contribution is a training scheme that implies using procedurally generated CAD models for SFT and handcrafted meshes for RL...**
>
> The reviewer recognized that with the previous authors' clarification on the experiment.
>
> **This is a valid concern. Yet, we only need this conversion for an ablation study...Actually, we cannot guarantee that the DeepCAD and CadQuery representations are 100% identical...Regarding the conversion procedure in Text-to-CadQuery. We find it extremely questionable,...**
>
> These are also the reasons I’m concerned about potential misalignments in comparisons. However, this issue arises from the pipelines used in other papers or projects, not from the core methodology of this work. And requiring 100% representational identity here would be overly stringent and impractical.
>
> So I think this slight misalignment does not harm the reflection on the main contribution.
>
> ---
>
> Congratulations on the improved clarity in your revision. All of my concerns have been satisfactorily resolved. And thorough experiments in this paper have been clarified to be exact and proper. The main contribution is clearly shown in the comprehensive ablation and is good to me, so I am raising my rating.

---

### Official Review · Reviewer_5cSp · 2025-10-30

**Soundness:** 3
**Presentation:** 3
**Contribution:** 3
**Rating:** 6
**Confidence:** 4

**Summary:**

This paper proposes cadrille, a multimodal framework for CAD reconstruction that unifies three input modalities (point clouds, multi-view images, and text descriptions) into a single generative model that outputs executable Python CAD code. cadrille combines SFT strategy on large-scale procedural CAD data with  RL fine-tuning on real handcrafted datasets using programmatic rewards. The proposed hybrid training pipeline significantly improves model validity and generalization across domains. Experiments on such as DeepCAD, Fusion360 dataset demonstrate consistent SOTA results across ten benchmarks.

**Strengths:**

1. The paper evaluates the proposed method on four different datasets (DeepCAD, Fusion360, CC3D, and Omni-CAD) and covers three input modalities. The reported performance improvements are significant and consistent, supporting the effectiveness of the proposed method.

2. The SFT (on procedural data) + RL (on handcrafted data) approach is a highly effective and novel solution to the well-known domain gap problem. Integrating LLM-style RL fine-tuning into multimodal CAD reconstruction is a novel and impactful idea.

3. Handling point clouds, images, and text within a single framework is technically elegant and practically valuable.

4. The paper includes excellent ablation analyses. It proves that its RL adaptation strategy is superior to a naive SFT on a mixed dataset

**Weaknesses:**

1. The model inherits high computational overhead from large VLMs, but efficiency metrics are not reported.

2. The SFT stage relies on the CAD-Recode dataset, which the authors note lacks certain operations (e.g., symmetric extrusions) found in DeepCAD. It is unclear if the RL stage can learn to generate new CAD operations it has never seen during SFT.

3. CAD-MLLM is supposed to  be compared and discussed in the experiment section, as this would more effectively demonstrate the superiority of the proposed model. Currently, its brief mention in the related work section is insufficient and appears too weak to prove the advanced performance of proposed model.

**Questions:**

1. Table 1 does not show the results of cadrille using only Rp. When cadrille is trained using only Rp, is it completely equivalent to CAD-Recode?

2. Regarding the results in the fourth row of Table 2, how is Rpi+Dpi mixed? Does it sample from both datasets with the same probability in each iteration, or just sequentially fine-tuned using Rpi first and then Dpi?

3. How sensitive is the RL fine-tuning performance to the weighting between the IoU and invalidity penalties in the reward function?

4. Table 1 does not show the results of cadrille using only Rp. When cadrille is trained using only Rp, is it completely equivalent to CAD-Recode?

---

> ### Author Response · Authors · 2025-11-17
>
> We thank the reviewer for the positive and detailed assessment, especially noting our multimodal coverage, the effectiveness of our SFT + RL pipeline, and the strength of our ablations.
>
> ---
> **Efficiency metrics are not reported.**
>
> Efficiency metrics are reported in Sec. B (Tab. 9; L. 881-889). As shown, multimodal $\texttt{cadrille}$ matches or slightly exceeds single-modality baselines in inference time despite supporting three modalities.
>
> ---
> **Can RL introduce new CAD operations missing from SFT?**
>
> No. RL sees only 3D meshes (Sec. 4.4; L. 250) and never receives new CAD codes, so it cannot introduce unseen CAD operations. Missing operators such as symmetric extrusion in DeepCAD are naturally decomposed into two standard extrusions in CadQuery syntax, so the expressivity gap is negligible.
>
> ---
> **CAD-MLLM is cited but the comparison is missing**
>
> Quantitative comparisons with CAD-MLLM appear in Sec. B (Tab. 4; L. 810-816) on Omni-CAD and Fusion360 (Tab. 6; L. 820-823) benchmarks. $\texttt{cadrille}$ consistently outperforms CAD-MLLM across all shared benchmarks.
>
> ---
> **Is SFT $\texttt{cadrille}$ trained only on $R_p$ equivalent to CAD-Recode?**
>
> Yes. As shown in the last rows of Tab. 1, adding modalities during SFT does not affect point cloud metrics. When trained on $R_p$ only, our SFT model is equivalent to CAD-Recode (same architecture and objective). The additional RL stage further improves over CAD-Recode; see Tab. 11.
>
> ---
> **How is $R_{pi}$ + $D_{pi}$ mixed (Tab. 2, row 4)?**
>
> We convert DeepCAD ($D$) programs into CadQuery and jointly train on the union with CAD-Recode ($R$). Because fully aligning the two syntaxes is difficult (L. 237), this naive mixture slightly hurts CD/IoU, despite improving IR. This supports the motivation of our methodology (Fig. 1a): avoid handcrafted CAD code and rely on meshes only during RL.
>
> ---
> **Ablation on IoU and invalidity penalties in the reward function.**
>
> We ablate the contribution of $r_\text{invalid}(\tau)$ below, while keeping the IoU reward fixed. We report IoU scores for image-based reconstruction on all three datasets after 3 epochs of RL fine-tuning (note that we shortened our default 20-epoch schedule for the purpose of the rebuttal). The results show no significant change in model performance for values around our default of -10 (L. 261).
>
> | $r_\text{invalid}$ | DeepCAD | Fusion360 | CC3D |
> |:-:|:-:|:-:|:-:|
> | -15 | 90.2 | 83.6 | 63.1 |
> | -10 | 90.6 | 83.3 | 62.5 |
> | -5 | 90.4 | 83.8 | 63.6 |

---

> > ### Comment · Reviewer_5cSp · 2025-11-26
> >
> > Thanks for your detailed response. All my concerns has been addressed and  I have updated my initial score.

---

### Official Review · Reviewer_9cAF · 2025-11-01

**Soundness:** 4
**Presentation:** 4
**Contribution:** 4
**Rating:** 8
**Confidence:** 4

**Summary:**

This paper presents Cadrille, a multimodal CAD reconstruction system that unifies three input modalities—point clouds, multi-view images, and text—to produce executable Python CAD programs. The model builds upon a vision-language backbone and adopts a two-stage training paradigm: (1) SFT on large-scale synthetic CAD data, followed by (2) RL fine-tuning using programmatic IoU- and validity-based feedback. Cadrille achieves state-of-the-art results on multiple benchmarks varied by datasets and experimental setups.

**Strengths:**

Cadrille is, to my knowledge, the first CAD framework that jointly handles point clouds, images, and text inputs in a single model while maintaining competitive performance with single-modal baselines.

The quantitative results are comprehensive and informative. The authors evaluated a wide range of existing methods to demonstrate the model’s advantages across them, including experiments on multiple datasets and under various setups.

The qualitative results further highlight the model’s contributions, showing clear improvements in generation quality. The selected examples are non-trivial and effectively illustrate the strengths of the proposed approach. Along with the quantitative results, this work presents strong empirical coverage on its effectiveness.

The paper carefully details the multimodal data generation pipeline, reward definition, and training setup, contributing valuable reproducibility and practical insight to the community.

**Weaknesses:**

While the authors stated that their two-stage training strategy is inspired by LLM pretraining, the core SFT + RL paradigm is nearly identical to previous works such as CADFusion [2] and CAD-Coder [1]. The main difference lies in applying this strategy across modalities rather than within a single one. This distinction should be discussed more explicitly; the claim of “first to use RL fine-tuning for multimodal CAD” is true, but the conceptual framework remains evolutionary rather than fundamentally new.

[1] CAD-Coder: An Open-Source Vision-Language Model for Computer-Aided Design Code Generation. ArXiv.

[2] Text-to-CAD Generation Through Infusing Visual Feedback in Large Language Models. ICML 2025.

**Questions:**

This seems to be primarily a naming issue: the term “multimodal CAD reconstruction” is somewhat confusing, as the tasks defined in this work do not involve reconstructing original CAD shapes. In my view, “multimodal CAD generation” would be a more accurate description.

Could the authors clarify the backbone model size and report ablations on model scaling to ensure fairness versus the baselines?

---

> ### Author Response · Authors · 2025-11-17
>
> We thank the reviewer for the positive assessment, especially noting that $\texttt{cadrille}$ is, to their knowledge, the first CAD framework to jointly support point clouds, images, and text within a single model while staying competitive with single-modal baselines. We also appreciate the reviewer’s comments on our carefully detailed SFT + RL setup and the strong empirical coverage provided by our experiments.
>
> ---
> **Positioning against single-modal CAD-Coder and Text-to-CAD methods.**
>
> As the reviewer notes, $\texttt{cadrille}$ supports three modalities yet already outperforms CAD-Coder and Text-to-CAD on the text-to-CAD task even without RL (Tab. 6). More importantly, our SFT + RL setup differs in a key way: prior works train both stages on the same handcrafted DeepCAD dataset, whereas $\texttt{cadrille}$ is the first to split the stages – using procedurally generated CAD code (CAD-Recode) for SFT and handcrafted meshes (DeepCAD) only for RL (Fig. 1a). This means our method never relies on handcrafted CAD code, which enables stronger cross-dataset and cross-modality generalization.
>
> ---
> **Could the authors report ablations on model size versus the baselines?**
>
> Model size and inference time ablations are provided in Sec. B (Tab. 9; L. 881–889). $\texttt{cadrille}$ uses the same LLM as CAD-Recode (Qwen2-1.5B) plus the smallest Qwen vision module (Qwen2-VL-2B) to support images. The ablations show that $\texttt{cadrille}$ matches or exceeds the efficiency of single-modal baselines while supporting all three modalities.

---

> > ### Comment · Reviewer_9cAF · 2025-11-26
> >
> > Thanks for the rebuttal. I am happy with the ablation study now.
> >
> > I would still recommend the authors reconsidering their naming of "CAD reconstruction" to their contribution, but this is a minor problem and does not affect my evaluation of the technical contribution of this paper.

---

### Official Review · Reviewer_vQB4 · 2025-11-01

**Soundness:** 3
**Presentation:** 2
**Contribution:** 3
**Rating:** 6
**Confidence:** 4

**Summary:**

This paper presents cadrille, a multimodal CAD reconstruction framework that unifies three input modalities (point clouds, multi-view images, and textual descriptions) to generate executable Python code for parametric CAD models. Built upon the Qwen2-VL vision-language model, the approach employs a two-stage training paradigm: (1) supervised fine-tuning (SFT) on large-scale procedurally generated data (from CAD-Recode), and (2) reinforcement learning (RL) fine-tuning on handcrafted data without requiring CAD sequence annotations. The method achieves state-of-the-art results across 10 benchmarks, with improvements in reducing invalidity ratios.

The key contributions are: (1) a multimodal CAD reconstruction method achieving SOTA performance, (2) a novel training strategy that separates large-scale synthetic data for SFT and smaller real-world data for annotation-free RL fine-tuning, and (3) comprehensive evaluation demonstrating robustness across synthetic and real-world scenarios.

**Strengths:**

1. SOTA: This work introduces a unified multimodal model for CAD reconstruction can match or exceed specialized single-modal approaches.
2. Training paradigm: The separation of procedurally generated data (SFT) and handcrafted data (RL) elegantly addresses the volume-quality tradeoff in CAD datasets. The key insight that RL can work without CAD sequence annotations (only requiring meshes) substantially lowers the barrier for future work.

**Weaknesses:**

1. Insufficient architectural innovation: The core contribution is training methodology rather than model design.

2. Critical phenomena lack adequate explanation: Cross-modal transfer (RL on images improves point clouds): The paper acknowledges this as "surprising" but provides no mechanistic explanation or controlled experiments. Possible hypotheses are not tested.

3. RL fine-tuning design choices lack justification.

**Questions:**

1. Cross-modal transfer mechanism: Can you explain why RL on images improves point cloud reconstruction? Does RL on point clouds also improve image reconstruction?

2. Dataset mixing failure (Table 3, row 4): The negative result when mixing Rpi+Dpi is attributed to "operation inconsistency", with the authors noting that "DeepCAD models are constructed using commands like extruded cuts and symmetric extrusions, which are not present in the generation procedure of the CAD-Recode dataset." However, this explanation raises questions: extruded cuts and extrusions are fundamental CAD operations that should theoretically be present in both datasets.

3. RL on point clouds: Why was RL fine-tuning not performed on point clouds? Why only images?

4. Hard example mining: R_th = 7.5 appears hand-tuned. How sensitive is performance to this threshold?

5. Can the author release the code for their paper？

---

> ### Author Response · Authors · 2025-11-17
>
> We thank the reviewer for the positive assessment, especially noting our unified multimodal framework and the two-stage SFT + RL training scheme that reduces dependence on large annotated datasets. We appreciate the recognition of our SOTA results and the contribution’s soundness.
>
> ---
> **Why was RL fine-tuning not performed on point clouds? Why only images?**
>
> As noted in L. 484, RL on point clouds is a promising future direction. In practice, RL on point clouds was unstable due to higher reward variance from sampling noise. In contrast, RL on images was stable and – importantly – already improves point cloud reconstruction (Tab. 3, rows 3 vs 6), suggesting the policy learns modality-shared CAD priors.
>
> ---
> **Both datasets in the mixture $D_{pi}$ and $R_{pi}$ should ideally have the same CAD operations.**
>
> We use DeepCAD ($D$) and CAD-Recode ($R$) datasets as provided. Although R is natively in CadQuery and we convert D to the same format, the underlying CAD operation vocabularies still differ (L. 238). For example, the symmetric extrusion mentioned by the reviewer is a single atomic operation in $D$ but corresponds to two explicit extrusions in $R$. Such mismatches are common across CAD datasets and even across CAD software. This is exactly why we do not mix CAD code across datasets: in our RL stage we mix only 3D meshes, avoiding operation-level inconsistencies while still benefiting from additional data.
>
> ---
> **Hard mining threshold $R_\text{th}=7.5$ appears hand-tuned.**
>
> Hard mining threshold $R_\text{th}$ is ablated in Sec. C (Tab. 12; L. 936-943)
>
> ---
> **Can the author release the code for their paper?**
>
> Yes. Our full codebase (SFT + RL + evaluation) is included in the Supplementary and will be made public upon publication (L. 491).

---

### Author Response · Authors · 2025-12-01
**Final authors' comment**

We thank all reviewers for the valuable discussion during the rebuttal period. The discussion focused primarily on clarifying implementation details of the Chamfer Distance evaluation and the comparison with CAD-MLLM. Following these exchanges, we introduced several minor revisions to the manuscript: (i) added an explicit description of the CD evaluation space and the lower bound across all datasets, (ii) clarified the comparison protocol with CAD-MLLM, including its training setup, and (iii) fixed some typos.

Initially, 4 out of 5 scores were already positive. The only negative score given by the Reviewer SjgS was updated after an intensive and fruitful discussion, which took place prior to the OpenReview leak. Overall, the scores were updated from (8, 2, 6, 8, 6) to **(8, 8, 8, 8, 6)**.

---

### Note · Program_Chairs · 2026-04-18
**Submission Desk Rejected by Program Chairs**

It has been brought to our attention that an arXiv submission with the same title, authors, and content (https://arxiv.org/pdf/2505.22914v1) as the ICLR submission indicates that the work was at least partially done at RAIRI, which is an institution sanctioned by the US. Upon review by the ICLR legal team, we are writing to inform you that the paper must be rejected in order to conform with US laws, where the ICLR organization is incorporated.

---

> ### Comment · Program_Chairs · 2026-04-18
>
> This decision is based on the list of sanctioned individuals and institutions [here] (https://sanctionslistservice.ofac.treas.gov/api/PublicationPreview/exports/SDN.CSV)

---

### Meta-Review · Area_Chair_EwEF · 2026-01-05

**Summary:**

This work focuses on Computer-Aided Design (CAD) tasks and proposes Cadrille, a multimodal CAD reconstruction framework that unifies three input modalities—point clouds, multi-view images, and textual descriptions—to generate executable Python code for parametric CAD models. Cadrille achieves state-of-the-art results across multiple benchmarks, varying in datasets and experimental setups. Reviewers raised concerns about the distinction of the SFT+RL method design, as well as the experimental setup and evaluation. The authors clarified misunderstandings and addressed most of these concerns. All reviewers support its acceptance.

**Reviewer Concerns:**

Concerns regarding method details, such as normalization and experimental results analysis, are resolved in the rebuttal. I would like to ask the authors to pay attention to clearly describing the motivation behind each technical design in cadrille, and also release the code.

**Reviewer Scores:**

The negatively scoring reviewers indicated a willingness to increase their scores.

---

### Decision · Program_Chairs · 2026-01-26

Accept (Oral)